# LEARNING TO REACH GOALS VIA DIFFUSION

## ABSTRACT

Diffusion models are a powerful class of generative models capable of mapping random noise in high-dimensional spaces to a target manifold through iterative denoising. In this work, we present a novel perspective on goal-conditioned reinforcement learning by framing it within the context of diffusion modeling. Analogous to the diffusion process, where Gaussian noise is used to create random trajectories that walk away from the data manifold, we construct trajectories that move away from potential goal states. We then learn a goal-conditioned policy analogous to the score function. This approach, which we call Merlin[1], can reach predefined or novel goals from an arbitrary initial state without learning a separate value function. We consider three choices for the noise model to replace Gaussian noise in diffusion - reverse play from the buffer, reverse dynamics model, and a novel non-parametric approach. We theoretically justify our approach and validate it on offline goal-reaching tasks. Empirical results are competitive with state-of-the-art methods, which suggests this perspective on diffusion for RL is a simple, scalable, and effective direction for sequential decision-making.

## 1 INTRODUCTION

Reinforcement learning (RL) has established itself as a powerful paradigm for agents to learn behaviors supervised by only a reward signal, demonstrating remarkable success across diverse applications ranging from robotics (Nguyen & La, 2019; Brunke et al., 2022) and game playing (Mnih et al., 2013; Silver et al., 2018) to recommendation systems (Zheng et al., 2018; Afsar et al., 2022) and autonomous driving (Liang et al., 2018; Kiran et al., 2021). Goal-conditioned RL (GCRL) aims to learn general policies that can reach arbitrary target states or goals within an environment without requiring extensive retraining (Kaelbling, 1993; Schaul et al., 2015; Chane-Sane et al., 2021). In GCRL, the task is defined in terms of a desired goal state, and the policy produces behavior that would ideally lead the agent to the specified goal. Despite its allure, training goal-conditioned RL agents presents inherent difficulties. Many GCRL tasks offer sparse rewards, necessitating intensive exploration, which can be infeasible and often unsafe in real-world scenarios. Concurrently, offline RL has gained attention in recent years for learning policies from fixed datasets, ensuring safety for real-world applications (Levine et al., 2020; Prudencio et al., 2023). Combining offline and goal-conditioned RL can potentially benefit from both generalization and data efficiency in practical scenarios.

However, offline goal-conditioned RL introduces new challenges. Many existing methods rely on learning a value function (Yang et al., 2021; Ma et al., 2022), which estimates the expected discounted return associated with a given state-action pair. During training, policies often generate actions not present in the offline dataset, leading to inaccuracies in value function estimation for out-of-distribution actions. These inaccuracies, compounded over time, can cause policies to diverge (Levine et al., 2020). The value estimation problem is further exacerbated by a sparse binary reward signal common in goal-conditioned settings. Prior attempts to tackle this issue involve constraints on policies or conservative value function updates (Kumar et al., 2019; 2020), which compromise policy performance (Levine et al., 2020) and make generalization challenging.

Another issue is that the dataset may only cover limited state-goal pairs out of the many possible combinations. Hindsight relabeling has been employed to generate positive goal-conditioned

---

[1]In Arthurian legends, Merlin the Wizard is said to have lived backward in time, perceiving events in reverse order, which gave him the ability to predict future events.

observations by replacing the desired goals with achieved goals appearing further along the same trajectory (Andrychowicz et al., 2017; Ghosh et al., 2020). However, hindsight relabeling can only generate state-goal pairs within the same trajectory, resulting in over-fitted goal-conditioned policies. Chebotar et al. (2021) proposed a goal-chaining technique that can generate state-goal pairs across the entire dataset, but this technique relies on a learned value function.

In light of these challenges, this paper draws inspiration from diffusion models — a powerful class of generative models that can map random noise in high-dimensional spaces to target manifolds through iterative denoising (Sohl-Dickstein et al., 2015; Ho et al., 2020). Building upon the idea of introducing controlled noise to destroy the structure of the target data distribution, we employ a similar strategy for goal-conditioned RL by constructing trajectories that move away from desired goals during the learning process. A goal-conditioned policy is then trained to reverse (or effectively "denoise") these trajectories. By navigating away from desired goals and subsequently correcting these deviations, the policy learns to reach any predefined goal state from arbitrary initial states. Notably, our approach, which we call Merlin, does not learn a value function. The resulting simple learning dynamics suggest the feasibility of stable and scalable training in the offline setting.

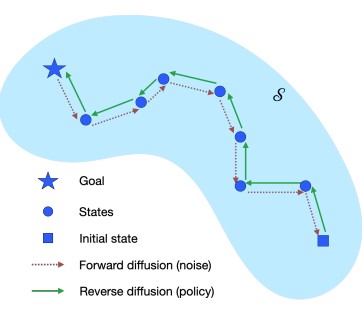

Figure 1: Reverse diffusion policy.

In terms of empirical evaluation, first, we substantiate Merlin in a simple yet illustrative point mass navigation environment, highlighting the parallels between score functions and the learned policy. We then discuss Merlin's applicability to offline datasets and provide theoretical justification for our approach. To this end, we discuss and empirically evaluate three possible choices for the forward diffusion noise model - reverse play using only the offline data, learning a reverse dynamics model, and a novel trajectory stitching technique grounded in nearest-neighbor search, which enables the generation of state-goal pairs *across* trajectories. The absence of a value function and the use of the trajectory stitching technique address two significant challenges encountered in offline GCRL.

Our contributions can be summarized as follows:

- Development of a novel goal-conditioned reinforcement learning approach inspired by diffusion models without learning a value function.
- Development of a novel trajectory stitching technique based on nearest-neighbor search that can generate state-goal pairs across trajectories.
- Demonstration of the effectiveness of our approach in various offline goal-reaching control tasks with significant performance enhancements compared to state-of-the-art methods.

## 2    RELATED WORK

**Offline Goal-Conditioned Reinforcement Learning.**    The core challenges faced by offline GCRL are the sparse binary nature of the rewards and limited coverage of the possible state-goal pairs within the offline dataset. Hindsight relabeling (HER) (Andrychowicz et al., 2017), a technique originally proposed for settings with online interactions, has been adapted by prior work to the offline setting. Ghosh et al. (2020) proposes a simple yet effective method that combines goal-conditioned behavior cloning with HER. Several other methods incorporate value learning methods and adapt them to the offline goal-conditioned setting. Yang et al. (2021) improves upon Ghosh et al. (2020) by incorporating discount-factor and advantage function weighting (Peng et al., 2019). Chebotar et al. (2021) generates state-goal pairs across trajectories using a goal-chaining technique that assigns conservative Q-values to out-of-distribution data. Ma et al. (2022) proposes an advantage-weighted regression approach with $f$-divergence regularization, which is based on state-occupancy matching. Notably, this approach does not use HER.

**Diffusion-based Reinforcement Learning.**    Recent works have leveraged diffusion models for offline RL by generating trajectory segments from random Gaussian noise. Janner et al. (2022) employs classifier-based guidance using a learned value function to guide the diffusion process to generate high-return trajectories. In contrast, Ajay et al. (2022) uses classifier-free guidance by

learning a denoising function conditioned on returns, goals, or constraints. These methods operate similarly to model predictive control (Garcia et al., 1989), where only the first action of the generated trajectory is performed. A distinct line of work represents the policy as a diffusion model where actions are sampled by denoising random Gaussian noise (Wang et al., 2022), conditioned on the states and guided by a learned value function. Both the trajectory and action sampling approaches can learn expressive policies but are computationally expensive due to the iterative denoising process required at each environment step. To our knowledge, Merlin is the first method that performs diffusion in the state space. States are diffused starting from potential goal states, and the policy is trained to reverse this diffusion, requiring only one iteration of "denoising" at each environment step. This distinction makes Merlin conceptually simpler and significantly more efficient than prior diffusion-based methods. As shown in our experiments, it also outperforms these methods in GCRL.

## 3 PRELIMINARIES

### 3.1 DIFFUSION PROBABILISTIC MODELS

Diffusion probabilistic models (Sohl-Dickstein et al., 2015; Ho et al., 2020) are generative models that can be used to model a data distribution. These latent variable models are characterized by a probability distribution that evolves over time, following a forward diffusion process. The forward diffusion process is generally fixed to add Gaussian noise at each timestep according to a variance schedule $\beta_1, \ldots, \beta_T$. Let $\mathbf{x}_0 \sim q(\mathbf{x}_0)$ denote the data and $\mathbf{x}_1, \ldots, \mathbf{x}_T$ denote the corresponding latent variables. The approximate posterior $q(\mathbf{x}_{1:T} \mid \mathbf{x}_0)$ is given by,

$$q(\mathbf{x}_{1:T}|\mathbf{x}_0) := \prod_{t=1}^{T} q(\mathbf{x}_t|\mathbf{x}_{t-1}), \qquad q(\mathbf{x}_t|\mathbf{x}_{t-1}) := \mathcal{N}(\mathbf{x}_t; \sqrt{1 - \beta_t}\mathbf{x}_{t-1}, \beta_t\mathbf{I}) \qquad (1)$$

Diffusion probabilistic models then learn a denoising function that reverses the forward diffusion process. Starting at $p(\mathbf{x}_T) = \mathcal{N}(\mathbf{x}_T; 0, \mathbf{I})$, the joint distribution of the reverse process is given by,

$$p_\theta(\mathbf{x}_{0:T}) := p(\mathbf{x}_T) \prod_{t=1}^{T} p_\theta(\mathbf{x}_{t-1}|\mathbf{x}_t), \qquad p_\theta(\mathbf{x}_{t-1}|\mathbf{x}_t) := \mathcal{N}(\mathbf{x}_{t-1}; \boldsymbol{\mu}_\theta(\mathbf{x}_t, t), \boldsymbol{\Sigma}_\theta(\mathbf{x}_t, t)) \qquad (2)$$

where $\boldsymbol{\mu}_\theta$ and $\boldsymbol{\Sigma}_\theta$ can be neural networks. The reverse process can produce samples matching the data distribution after a finite number of transition steps. Note that traditionally, $t = 0$ corresponds to the data and higher timesteps correspond to noisy latent variables. In our discussion, we set the goal at the maximum timestep $T$, and decrease the timestep during forward diffusion to keep the timestep consistent with standard RL notation.

### 3.2 GOAL-CONDITIONED REINFORCEMENT LEARNING

The RL problem can be described using a Markov Decision Process (MDP), denoted by a tuple $(\mathcal{S}, \mathcal{A}, \mathcal{P}, r, \mu, \gamma)$, where $\mathcal{S}$ and $\mathcal{A}$ are the state and action spaces; $\mathcal{P}$ describes the transition probability as $\mathcal{S} \times \mathcal{A} \times \mathcal{S} \to [0, 1]$, $r : \mathcal{S} \times \mathcal{A} \to \mathbb{R}$ is the reward function, $\mu(s)$ is the initial state distribution, and $\gamma \in (0, 1]$ is the discount factor.

Goal-conditioned RL additionally considers a goal space $\mathcal{G} := \{\phi(s) \mid s \in \mathcal{S}\}$, where $\phi : \mathcal{S} \to \mathcal{G}$ is a known state-to-goal mapping (Andrychowicz et al., 2017). For example, in the FetchPush task, a robotic arm is tasked with pushing a block to a goal position specified as $(x, y, z)$ coordinates of the block, but the state represents the positions and velocities of the various effectors and components of the robotic arm as well as the block. The reward function now depends on the goal, $r : \mathcal{S} \times \mathcal{A} \times \mathcal{G} \to \mathbb{R}$. Generally, the reward function is sparse and binary defined as $r(s, a, g) = \mathbb{1}[\|\phi(s) - g\|_2^2 \leq \delta]$, where $\delta$ is some threshold distance.

A goal-conditioned policy is denoted by $\pi : \mathcal{S} \times \mathcal{G} \to \mathcal{A}$, and given a distribution over desired goals $p(g)$, an optimal policy $\pi^*$ aims to maximize the expected return,

$$J(\pi) = \mathbb{E}_{g \sim p(g), s_0 \sim \mu(s_0), a_t \sim \pi(\cdot|s_t, g), s_{t+1} \sim \mathcal{P}(\cdot|s_t, a_t)} \left[ \sum_{t=0}^{\infty} \gamma^t r(s_t, a_t, g) \right],$$

For offline RL problems, the agent cannot interact with the environment during training and only has access to a static dataset $D = \{(s_t, a_t, g, r_t)\}$, which can be collected by some unknown policy.

## 4 REACHING GOALS VIA DIFFUSION

Consider the generative modeling problem of generating samples from some distribution $p_{\text{data}}(\mathbf{x})$ given a set of samples $\{\mathbf{x}_i\}_{i=1}^N, \mathbf{x}_i \in \mathbb{R}^d$. Diffusion modeling entails constructing a Markov chain which iteratively adds Gaussian noise to these samples. The perturbed data effectively covers the space surrounding the data manifold. The learned reverse diffusion process can then map any point drawn from a Gaussian distribution in $\mathbb{R}^d$ to a point on the data manifold.

Now consider a goal-augmented MDP $(\mathcal{S}, \mathcal{A}, \mathcal{G}, \mathcal{P}, r, \mu, \gamma)$ with goals $g \in \mathcal{G}$ sampled from some unknown goal distribution $g \sim p(g)$. Goal-conditioned RL aims to learn a policy that can learn an optimal path from any state $s \in \mathcal{S}$ to the desired goal $g$. This can be viewed as similar to learning to map random noise in $\mathbb{R}^d$ to some target data manifold, except that the underlying space is now restricted to the state space of the MDP.

Learning to reach goals using diffusion requires constructing a forward diffusion process and a corresponding reverse process. In the context of RL, the forward process comprises taking actions starting from the goal, which can be seen as equivalent to adding Gaussian noise in diffusion models, which leads to new states different from the goal. We apply this process iteratively for $T$ steps, where $T$ denotes the maximum length of the diffusion chain. We then train a policy conditioned on the original goal state to reverse this trajectory.

### 4.1 AN ILLUSTRATIVE EXAMPLE

We illustrate this concept using a simple 2D navigation environment consisting of an agent tasked with reaching a target goal state. The states are the $(x, y)$ coordinates of the agent and actions represent the displacement in the $x$ and $y$ directions, normalized to be unit length. For these experiments, the goal state during training is fixed to $g = (0, 0)$, and the initial agent state is sampled uniformly at random. For this simple environment, given a state $s_t$ and an action $a_{t-1}$, it is straightforward to compute the previous state $s_{t-1}$ such that taking action $a_{t-1}$ at $s_{t-1}$ would lead to $s_t$.

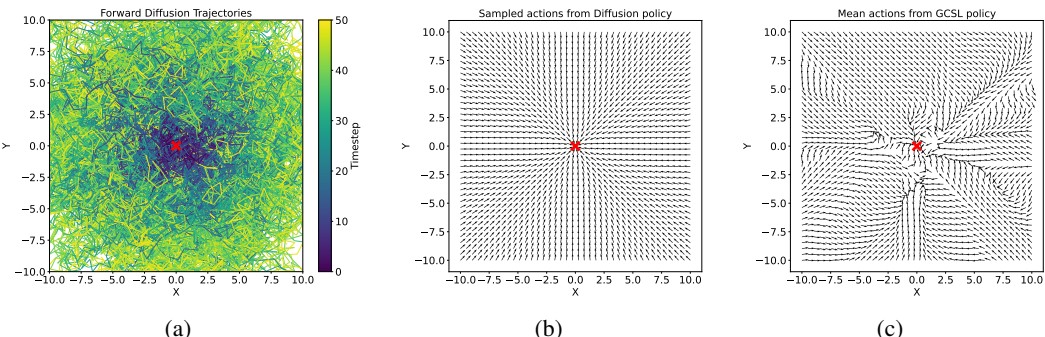

| (a) | (b) | (c) |

Figure 2: (a) Visualization of trajectories starting from the goal **X** generated during the forward process, (b) Predicted actions from policy trained via diffusion, (c) Predicted actions from policy trained using GCSL.

Figure 2a visualizes the trajectories representing the forward diffusion process, obtained by taking random actions starting from the goal for $T = 50$ steps. We set $s_T = g$ and the random state reached at the end of the diffusion is $s_1$. The policy is parameterized as a diagonal Gaussian distribution and is trained to reverse these trajectories by conditioning on the final goal or some future state in the trajectory. More formally, for a trajectory $\tau = \{s_1, a_1, \ldots, s_{T-1}, a_{T-1}, s_T = g\}$, the policy parameters are trained by optimizing $\theta^* = \arg\max_\theta \mathbb{E}_{(s_t, a_t, g') \sim \tau} \log \pi_\theta(a_t | s_t, g', h)$ where $g' = \phi(s_i)$ and $h = i - t$ for $t < i \le T$. In words, for any state $s_t$ in the trajectory, we maximize the likelihood of the observed action $a_t$, conditioned on any future (goal) state $g'$, given the time horizon $h$ separating the two states in the observed trajectory.

Figure 2b visualizes the actions sampled from the trained policy for different states when conditioned on the goal $g = (0, 0)$ using an input time horizon of one. For comparison, Figure 2c visualizes the trained policy using GCSL (Ghosh et al., 2020). Both methods were trained for $100k$ policy updates. The policy learned via diffusion learns the optimal path, which takes the shortest time to reach the goal. We extend this example to more complex settings in Appendix B. It is interesting to note that



| (a) h=1 | (b) h=5 | (c) h=10 | (d) h=20 | (e) h=50 |
|---------|---------|----------|----------|----------|

Figure 3: Actions sampled from the trained policy, showcasing the effect of time horizon during evaluation.

the trained policy is analogous to the score function for diffusion models, and the action is analogous to the predicted noise, which serves to "denoise" the states towards the goal. We elaborate on this relation to diffusion probabilistic models in Appendix C.

For policy evaluation, the choice of time horizon to be used is not immediately obvious. We investigate the effect of changing the time horizon which is shown in Figure 3. For $h = 1$, the policy always takes the most direct path to the goal regardless of the input state. For larger values of the time horizon, the policy has a high variance close to the goal and a low variance for the optimal action further away. In Section 6.1, we perform ablations to further investigate its effect on performance.

We then test the generalization capabilities of our approach by evaluating the policy on out-of-distribution goals. During training, the goal is fixed to be at the center but during evaluation, we condition the policy on random goals. Figure 4 shows that the policy can effectively generalize to different goals due to hindsight relabeling. Note that GCSL also uses hindsight relabeling, but unlike Merlin, it often takes sub-optimal paths to reach the goal.

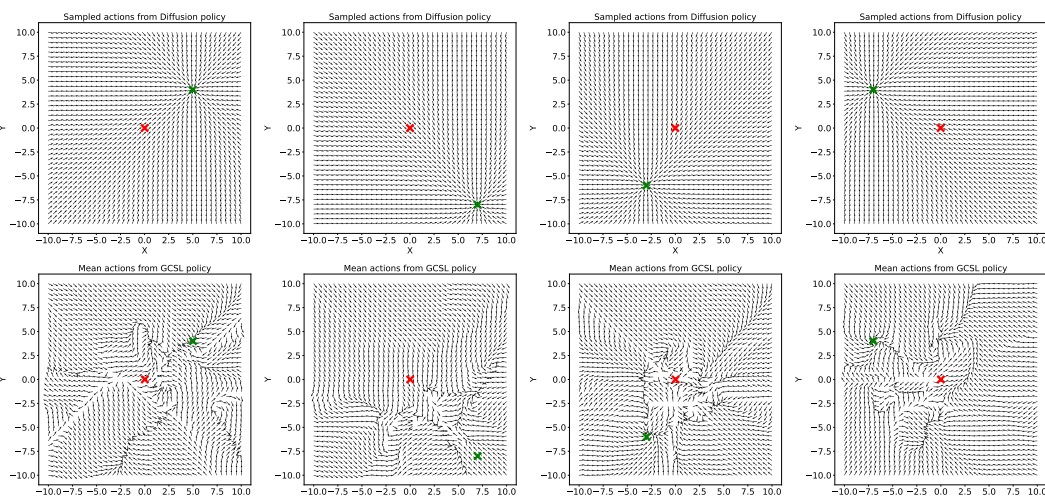

Figure 4: Evaluating the trained policy on out-of-distribution goals. Red **X** denotes the goal used during training, and green **X** denotes the goal used for evaluation. **Top:** Diffusion; **Bottom:** GCSL.

## 5 GOAL-CONDITIONED DIFFUSION POLICY

Section 4 suggests learning policies by constructing diffusion chains and reversing them can be quite effective. In this section, we discuss the application of Merlin to offline datasets.

A straightforward application of Merlin to offline data involves loading the dataset to a replay buffer and sampling trajectories for reverse play. To create more varied diffusion trajectories beyond the dataset, we can learn a reverse dynamics model and optionally even a separate reverse policy (Wang et al., 2021). Like most model-based methods, we found this method suffers from compounding errors over multiple time steps, resulting in poor quality data and unstable policy training. We, therefore, introduce a novel non-parametric method that stitches trajectories to create diverse diffusion paths. Once we have specified the procedure for producing reverse trajectories, we discuss the procedure for training a goal-conditioned policy and provide theoretical guarantees.

**Algorithm 1** Nearest-neighbor Trajectory Stitching

**Input:** Dataset $\mathcal{D}$, distance threshold $\delta$, number of new trajectories to collect $M$
**Output:** Augmented dataset $\mathcal{D}_{\text{new}}$
$\mathcal{D}_{\text{new}} \leftarrow \mathcal{D}$
Construct ball tree $T$ for all states
**for** $m \leftarrow 1, \dots, M$ **do**
    Sample random final state $s_T$ from $\mathcal{D}$
    $\tau_{\text{new}} \leftarrow \{s_T\}$
    $s_{\text{current}} \leftarrow s_T$
    **for** $t \leftarrow T, \dots, 1$ **do**
        $s_{\text{neighbor}}, d \leftarrow T.\text{query}(s_{\text{current}}, k = 1)$
        **if** $d \leq \delta$ **then**
            Add preceding $(s_{\text{prev}}, a_{\text{prev}})$ from $s_{\text{neighbor}}$ to $\tau_{\text{new}}$
        **else**
            Add preceding $(s_{\text{prev}}, a_{\text{prev}})$ from $s_{\text{current}}$ to $\tau_{\text{new}}$
        **end if**
        $s_{\text{current}} \leftarrow s_{\text{prev}}$
    **end for**
    $\mathcal{D}_{\text{new}} \leftarrow \mathcal{D}_{\text{new}} \cup \tau_{\text{new}}$
**end for**
**Return:** $\mathcal{D}_{\text{new}}$

Figure 5: Trajectory stitching.

## 5.1 Nearest-neighbor Trajectory Stitching

The forward diffusion constructs trajectories walking away from the goal to provide training data for the policy. In order for this strategy to be effective, we want to generate as many state-goal pairs as possible to help the policy generalize well. Hindsight relabeling can generate positive goal-conditioned observations by replacing the desired goals with achieved goals appearing further along the *same* trajectory.

We introduce a novel trajectory stitching operation to generate state-goal pairs *across* trajectories. The basic idea behind this operation is that if two states from different trajectories are very close to each other, then the sub-trajectory leading up to one of these states is also a reasonably good path to reach the other state.

The formalize this notion, we have to choose a metric and corresponding threshold. We choose the Euclidean distance, although other valid metrics may also be used. The distance threshold varies for different tasks, as it depends on the state dimension and the properties of the underlying state space. We construct a ball tree for all the states in the dataset to allow quick nearest-neighbor search. Note that for a $d$-dimensional dataset with $N$ samples, the query time for the ball tree has time complexity $O(d \log N)$. We sample random goal states from the dataset and iteratively add the previous state-action to the new trajectories. At each step, we query the ball tree for the nearest neighbor and if the nearest distance is less than the threshold, we switch to the trajectory corresponding to the neighbor state, otherwise, we stick to the same trajectory. Algorithm 1 presents this procedure. The choice of the metric and the value of $\delta$ is discussed in Appendix D.4.

## 5.2 Offline Policy Training

Consider a fixed dataset of trajectories $\mathcal{D}$ generated by some unknown behavior policy $\pi_\beta$, and a trajectory $\tau \sim \mathcal{D}$, where $\tau = \{s_1, a_1, \dots, s_T\}$. We can view this trajectory in reverse – starting from the final state $s_T$, we apply an unknown transformation to each state $s_{t+1}$ to obtain state $s_t$. The corresponding forward diffusion process is denoted by $q(s_t|s_{t+1})$. We train a policy denoted by $\pi_\theta$ to reverse this diffusion. The corresponding reverse diffusion process is given by $p_\theta(s_{t+1}|s_t) = \mathcal{P}(s_{t+1}|s_t, \pi_\theta(\cdot|s_t, g))$, where $g = \phi(s_T)$ is the goal. Our objective is to maximize the log-likelihood of the goal states under the reverse diffusion process.

**Theorem 5.1.** *Consider a dataset $\mathcal{D}(g)$ collected by an unknown behavior policy $\pi_\beta$, consisting of trajectories ending in states $S_T := \{s_T \mid g = \phi(s_T)\}$ with $q(s_T|g)$ denoting the distribution of final states corresponding to $g$. Then, behavior cloning given by $\theta^* = \arg\max_\theta \mathbb{E}_{(s,a)\sim\mathcal{D}(g)}\left[\log \pi_\theta(a|s)\right]$ is equivalent to maximizing a lower bound on the log-likelihood of the final states under the reverse diffusion process $L = \mathbb{E}_{s_T\sim q(s_T|g)}\left[\log p_\theta(s_T)\right]$.*

The proof is provided in Appendix A. Suppose we sample different datasets $\mathcal{D}(g)$ for different goals $g \sim p(g)$, where $\mathcal{D}(g)$ is produced from dataset $\mathcal{D}$ using hindsight relabeling. Additionally, we condition the policy on the goal $g$. Repeated application of the theorem above for $g \sim p(g)$ results in the following corollary.

**Corollary 5.1.** *Given a dataset of trajectories $\mathcal{D}$ and target goal distribution $p(g)$, behavior cloning using a goal-conditioned policy $\theta^* = \arg\max_\theta \mathbb{E}_{g\sim p(g),(s,a)\sim\mathcal{D}(g)}\left[\log \pi_\theta(a|s,g)\right]$ maximizes a lower bound on the log-likelihood of the goal states $L = \mathbb{E}_{g\sim p(g),s_T\sim q(s_T|g)}\left[\log p_\theta(s_T)\right]$.*

Similar to denoising diffusion models, we additionally condition the policy on the time horizon $h$ separating the current state and the goal state. In our experiments, the policy is parameterized as a diagonal Gaussian distribution,

$$\pi_\theta(\cdot|s_t,g,h) = \mathcal{N}(\cdot|\mu_\theta(s_t,g,h), \sigma_\theta^2(s_t,g,h)\mathbf{I})$$

Note that most prior works that employ behavior cloning do not learn the variance term and minimize the mean squared error between observed and predicted actions. As seen in Section 4, incorporating a learned variance allows the policy to incorporate uncertainty in their action predictions. This is important in learning from a trajectory far from the goal state.

In practice, we apply the trajectory stitching operation described in Section 5.1 along with hindsight relabeling to generate an augmented dataset $\mathcal{D}_{\text{new}}$ consisting of transitions $\{s_t, a_t, g = \phi(s_{t+h}), h\}$, where $h > 0$ is the horizon such that $t + h \leq T$, and $s_{t+h}$ is the hindsight labeled future state, possibly from a stitched trajectory. Following Theorem 5.1, the policy is trained using behavior cloning to maximize the log probability of actions given by these transitions,

$$\theta^* = \arg\max_\theta \mathbb{E}_{(s_t,a_t,g,h)\sim\mathcal{D}_{\text{new}}}\left[\log \pi_\theta(a_t|s_t,g,h)\right] \tag{3}$$

## 6 EXPERIMENTS

**Tasks.** We evaluate Merlin on several goal-conditioned control tasks using the benchmark introduced in Yang et al. (2021). The benchmark consists of 10 different tasks of varying difficulty, with the maximum trajectory length fixed to be $T = 50$ for all tasks. The tasks include the relatively easier PointReach, PointRooms, Reacher, SawyerReach, SawyerDoor, and FetchReach tasks with 2000 trajectories each ($1 \times 10^5$ transitions); and the harder tasks include FetchPush, FetchPick, FetchSlide, and HandReach with 40000 trajectories ($2 \times 10^6$ transitions). The benchmark consists of two settings 'expert' and 'random'. The 'expert' dataset consists of trajectories collected by a policy trained using online DDQPG+HER with added Gaussian noise ($\sigma = 0.2$) to increase diversity, while the 'random' dataset consists of trajectories collected by sampling random actions. The dataset includes the desired goal for each trajectory in addition to the state-action pairs. The reward for each task is sparse and binary, $+1$ for reaching the goal, and $0$ everywhere else. Further details on these tasks are provided in Appendix F.

**Algorithms.** We compare with state-of-the-art offline GCRL methods, as well as recently proposed diffusion-based RL methods. The GCRL methods are: (1) GCSL (Ghosh et al., 2020) which uses behavior cloning with hindsight relabeling, (2) WGCSL (Yang et al., 2021) that improves upon GCSL by incorporating discount factor and advantage function weighting, (3) ActionableModel (AM) (Chebotar et al., 2021) which uses an actor-critic method with conservative Q-learning and goal chaining, and (4) GoFAR (Ma et al., 2022) which uses advantage-weighted regression with $f$-divergence regularization based on state-occupancy matching. The diffusion-based methods are: (1) Decision Diffuser (DD) (Ajay et al., 2022) which generates full trajectories from random Gaussian noise using (in our case, goal) conditional diffusion for classifier-free guidance, and (2) Diffusion-QL (DQL) (Wang et al., 2022) that represents the policy as a diffusion model to sample actions, guided by a learned value function. We implement a goal-conditioned version (g-DQL) by additionally conditioning the policy on goals. Appendix E provides implementation details of the baselines.

We implement three variations of Merlin, all of which use behavior cloning and hindsight relabeling,

- **Merlin** uses the offline data loaded to a replay buffer, and samples trajectories for reverse play.
- **Merlin-P** uses a learned parametric reverse dynamics model and reverse policy as proposed in Wang et al. (2021) to generate additional diffusion trajectories starting from goal states, in addition to the original offline data.
- **Merlin-NP** uses the non-parametric trajectory stitching method introduced in Section 5.1 to generate diverse diffusion trajectories, in addition to the original offline data.

We train each method for $500k$ policy updates using a batch size of $512$, and the results are averaged over 10 seeds. Complete architecture and hyperparameter values as well as additional implementation details for the three variants of Merlin are provided in Appendix D. We tune two hyperparameters - the hindsight relabeling ratio and the time horizon on each individual task. For the baselines, we use the best reported hyperparameter values.

**Experimental Results.** Table 1 presents the discounted returns using the sparse binary task reward. This metric takes into account how fast the agent reaches the goal and whether it stays in the goal region thereafter. We also report the final success rate in Appendix H. The results show that the basic implementation of Merlin performs better than the baselines on most tasks. Merlin-NP improves the performance further achieving the highest discounted returns on most tasks, and is overall the best-performing method. Merlin-P performs well on the easier tasks, however, on tasks with relatively high state dimensions, the compounding model error leads to a substantial drop in performance. Since Merlin does not perform multiple denoising steps for each environment step, training and inference are roughly an order of magnitude faster than the other diffusion-based methods (DD and g-DQL). We compare training and inference times for all methods in Appendix D.4.

Both Merlin and GCSL employ behavior cloning with hindsight relabeling, with two key differences: (1) Merlin learns the variance of the policy in addition to the mean, which provides additional flexibility during optimization, and (2) Merlin conditions the policy on the time horizon similar to the denoising function in diffusion models. GCSL also allows for this conditioning, however, it does not learn the variance and reports similar performance with and without conditioning on the time horizon. Figure 3 illustrates the effect of time horizon on the learned variance, and the section below further demonstrates its effect on the performance of Merlin. Beyond the vanilla setting of reverse play from the buffer, the forward view of GCSL and the backward view of Merlin, which is inspired by diffusion, can result in very different outcomes. Consider the model-based approach: a forward dynamics model generates trajectories without guarantees on the distribution over the goal state. In contrast, in a reverse dynamics model, one has control over this distribution.

Table 1: Discounted returns, averaged over 10 seeds.

| | Task Name | Ours | | | Offline GCRL | | | | Diffusion-based | |
|---|---|---|---|---|---|---|---|---|---|---|
| | | **Merlin** | **Merlin-P** | **Merlin-NP** | **GoFAR** | **WGCSL** | **GCSL** | **AM** | **DD** | **g-DQL** |
| **Expert** | PointReach | **29.26**±0.04 | 29.17±0.15 | **29.30**±0.05 | 27.18±0.65 | 25.91±0.87 | 22.85±1.26 | 26.14±1.11 | 10.03±0.88 | 28.65±0.44 |
| | PointRooms | 25.38±0.37 | 25.25±0.07 | **25.42**±0.32 | 20.40±1.00 | 19.90±0.99 | 18.28±2.29 | 23.24±1.58 | 5.84±2.67 | **27.53**±0.57 |
| | Reacher | 22.75±0.59 | 23.25±0.17 | **24.97**±0.54 | 22.51±0.82 | **23.35**±0.64 | 20.05±1.37 | 22.36±1.03 | 4.39±1.08 | 22.54±1.42 |
| | SawyerReach | 26.89±0.07 | 25.05±0.60 | **27.35**±0.06 | 22.82±1.15 | 22.07±1.46 | 19.20±1.79 | 23.56±0.33 | 3.39±0.75 | 24.17±0.01 |
| | SawyerDoor | 26.18±2.19 | 25.75±0.97 | 26.15±2.08 | 23.62±0.35 | 23.92±1.10 | 20.12±1.33 | **26.39**±0.42 | 7.85±0.77 | 24.81±0.38 |
| | FetchReach | 30.29±0.03 | 30.26±0.02 | **30.42**±0.04 | 29.21±0.26 | 28.17±0.38 | 23.68±1.07 | 29.08±0.12 | 1.55±0.68 | 28.71±0.15 |
| | FetchPush | 19.91±1.20 | 2.23±2.20 | 21.58±1.63 | **22.41**±1.69 | **22.22**±1.51 | 17.58±1.47 | 19.86±3.16 | 5.49±2.85 | 17.82±0.55 |
| | FetchPick | 19.66±0.78 | 1.43±1.01 | **20.41**±0.92 | **19.79**±1.12 | 18.32±1.56 | 12.95±1.90 | 17.04±3.81 | 2.76±0.64 | 14.45±0.62 |
| | FetchSlide | 4.19±1.89 | 0.00±0.00 | **4.95**±2.02 | 3.34±1.01 | **5.17**±3.17 | 1.67±1.41 | 3.31±1.46 | 1.21±0.59 | 0.98±0.59 |
| | HandReach | 22.11±0.55 | 0.00±0.00 | **24.93**±0.49 | 15.39±6.37 | 18.05±5.12 | 0.15±0.11 | 0.00±0.00 | 0.00±0.00 | 0.00±0.00 |
| | Average Rank | **2.7** | 5.3 | **1.6** | 4.5 | 4.6 | 7.3 | 5.0 | 8.3 | 5.1 |
| **Random** | PointReach | **29.26**±0.04 | 29.21±0.08 | **29.31**±0.04 | 23.96±0.93 | 25.76±0.96 | 17.74±1.84 | 25.55±0.57 | 10.12±0.72 | 22.65±1.57 |
| | PointRooms | **24.80**±0.36 | 24.07±0.19 | **25.16**±0.59 | 18.09±4.13 | 19.41±1.01 | 14.69±2.51 | 19.10±1.39 | 5.76±2.99 | 20.88±0.96 |
| | Reacher | 21.09±0.65 | 16.65±0.48 | 22.24±0.54 | **25.10**±0.68 | 22.98±0.91 | 10.62±2.30 | **23.70**±0.62 | 4.74±0.36 | 6.06±0.84 |
| | SawyerReach | 26.70±0.14 | 25.46±0.12 | **26.86**±0.07 | 19.48±1.39 | 21.32±1.40 | 8.78±2.59 | 25.29±0.15 | 3.46±0.86 | 2.84±0.05 |
| | SawyerDoor | 19.05±0.66 | 18.26±1.18 | **21.69**±2.36 | **20.69**±2.14 | 19.58±3.55 | 12.47±3.08 | 10.82±1.67 | 7.92±0.86 | 14.77±0.51 |
| | FetchReach | 30.42±0.04 | 30.38±0.02 | **30.42**±0.04 | 28.34±0.98 | 27.94±0.30 | 18.96±1.77 | 27.11±0.22 | 1.71±0.77 | 1.21±0.46 |
| | FetchPush | 5.21±0.43 | 5.08±0.32 | **7.22**±0.35 | **6.99**±1.27 | 5.35±3.36 | 4.22±2.19 | 4.53±1.94 | 4.49±1.34 | 5.35±0.23 |
| | FetchPick | 3.75±0.18 | 3.02±0.16 | **4.36**±0.19 | **3.81**±3.71 | 1.87±1.59 | 0.81±0.82 | 3.08±1.35 | 2.16±0.75 | 2.17±0.18 |
| | FetchSlide | **2.67**±0.35 | 0.00±0.00 | **3.15**±0.14 | 1.32±1.22 | 1.04±0.98 | 0.24±0.27 | 1.12±0.39 | 1.31±0.52 | 0.00±0.00 |
| | HandReach | 14.89±2.54 | 0.00±0.00 | **17.61**±3.06 | 0.08±0.07 | 2.54±1.42 | 1.41±0.51 | 0.00±0.00 | 0.00±0.00 | 0.00±0.00 |
| | Average Rank | **2.8** | 4.8 | **1.3** | 3.8 | 4.5 | 7.3 | 5.3 | 7.7 | 6.7 |

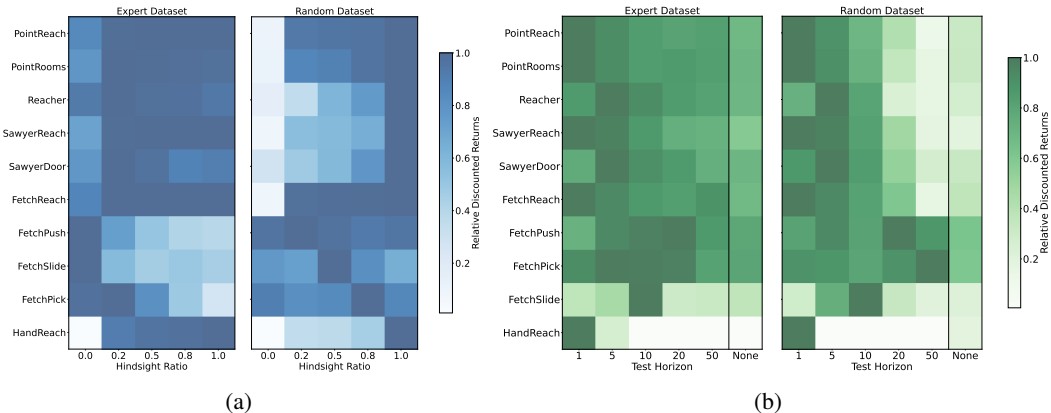

Figure 6: Discounted returns for each dataset with different values of (a) hindsight ratio and (b) time horizon during evaluation. Values are normalized with respect to the maximum value in each row.

## 6.1 ABLATION STUDIES

**Hindsight Relabeling.** During training, we employ hindsight relabeling to replace the desired goals with a future state further along the trajectory. A hyperparameter, which we call the *hindsight ratio*, specifies the fraction of each sampled batch of transitions that are subjected to this relabeling operation. As shown in Figure 6a, this ratio can significantly affect performance depending on the dataset. In general, a low-to-moderate value for the expert datasets and a high value for the random datasets seem to result in good performance. This observation can be explained by the fact that a large number of the expert trajectories reach the desired goals hence the original state-goal pairs provide good quality data for training the policy. On the other hand, the random trajectories benefit more from hindsight relabeling since state-goal pairs in the original dataset are sub-optimal, and relabeling provides realistic state-goal pairs to the policy. For the baselines that use hindsight relabeling, we use the optimal hindsight ratio as reported in their works.

**Time Horizon.** During training, the time horizon indicates the time difference between the current and desired goal states. During evaluation, the optimal value of the time horizon depends on the environment, as shown in Figure 6b. The last column, labeled 'None' shows the performance without conditioning on the time horizon, and for all tasks conditioning on the time horizon performs much better than without. For the easier tasks, a time horizon of $h = 1$ or $h = 5$ seems to work best, whereas for the more complex tasks, a higher value seems optimal. This can be attributed to the fact that for the more difficult tasks, the policy is expected to require more time steps to successfully reach the goal. In particular, the HandReach task seems especially sensitive to the time horizon, as using $h = 1$ performs significantly better than other values or without using time horizon conditioning. The optimal values for the hindsight ratio and the time horizon are provided in Appendix D.2.

## 7 CONCLUSION

We introduce Merlin, a goal-conditioned reinforcement learning method that draws inspiration from generative diffusion models. Distinct from other works that use diffusion for RL, we construct trajectories that "diffuse away" from potential goals and train a policy to reverse them, analogous to the score function. We discuss several choices to construct the forward diffusion process and introduce a novel trajectory stitching method that can be applied to most offline RL algorithms. We evaluate Merlin on various offline goal-conditioned control tasks and demonstrate superior performance compared to prior works. While Merlin is simple and performant, it is not without limitations. The trajectory stitching operation requires an explicit metric that may not be obvious in complex state spaces (such as images), which can be addressed by performing this operation in a learned latent space. Moreover, the interesting setting, which uses a reverse dynamics model for diffusion, suffers from a problem common to model-based techniques. We plan to further investigate the model-based approach in future work. While our work focused on the offline setting, extending Merlin to the online setting presents another interesting avenue for future work.

## 8 REPRODUCIBILITY STATEMENT

We have made efforts to ensure that our work is reproducible. Algorithm 1 provides a formal procedure for the nearest-neighbor trajectory stitching technique. Section 6 provides relevant experimental details including ablation studies on the two hyperparameters - the hindsight ratio and the time horizon. Appendix D provides implementation details for all three variations of Merlin, including network architectures, hyperparameters, and additional discussion on the trajectory stitching method. Appendix E provides a detailed description of all baselines considered in our experiments, along with links to their publicly available implementations that were used to produce the results. Appendix F describes the tasks that were used to evaluate Merlin. We plan to make the code publicly available after the reviewing process.

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

## A  PROOF OF THEOREM 5.1

**Setting.**  Consider a dataset $\mathcal{D}(g)$ where each trajectory is of the form $\tau = \{s_1, a_1, \ldots, s_T\}$ generated by some unknown behavior policy $\pi_\beta$. The final states are such that $g = \phi(s_T)$. We view these trajectories in reverse - starting from the final state $s_T$, we apply some unknown transformation to state $s_{t+1}$ to obtain state $s_t$. The corresponding forward diffusion process is denoted by $q(s_t|s_{t+1})$.

**Outline.**  The basic steps involved in the proof are:

1. Define the forward and reverse diffusion processes.
2. Obtain the distribution of final states achieved by the reverse diffusion process.
3. Define the log-likelihood of final states under the reverse diffusion process. Lower bound the log-likelihood using Jensen's inequality and simplify the resulting expression.
4. Obtain the optimal policy parameters by maximizing the lower bound for (a) deterministic and (b) stochastic MDPs.

**Proof.**

1. Let $q(s_T|g)$ denote the target distribution of final states corresponding to the goal $g$. For brevity, we denote it simply as $q(s_T)$, since in this setting the goal $g$ is fixed. The forward diffusion trajectory, starting at $s_T$ and performing $T$ steps of diffusion is thus,

$$q(s_1, \ldots, s_T) = q(s_T) \prod_{t=1}^{T-1} q(s_t|s_{t+1}),$$

   We train a policy denoted by $\pi_\theta(\cdot|s_t)$ to reverse this diffusion. The corresponding reverse diffusion process is given by,

$$p_\theta(s_{t+1}|s_t) = \mathcal{P}(s_{t+1}|s_t, \pi_\theta(\cdot|s_t)),$$

   The generative process corresponding to this reverse diffusion is,

$$p_\theta(s_1, \ldots, s_T) = p(s_1) \prod_{t=1}^{T-1} p_\theta(s_{t+1}|s_t),$$

   where $p(s_1)$ is the distribution of initial states.

2. The distribution of final states achieved by the reverse diffusion process,

$$
\begin{aligned}
p_\theta(s_T) &= \int ds_1 \ldots ds_{T-1} \, p_\theta(s_1, \ldots, s_T) \\
&= \int ds_1 \ldots ds_{T-1} \, q(s_1, \ldots, s_{T-1}|s_T) \frac{p_\theta(s_1, \ldots, s_T)}{q(s_1, \ldots, s_{T-1}|s_T)} \\
&= \int ds_1 \ldots ds_{T-1} \, q(s_1, \ldots, s_{T-1}|s_T) p(s_1) \prod_{t=1}^{T-1} \frac{p_\theta(s_{t+1}|s_t)}{q(s_t|s_{t+1})}
\end{aligned}
$$

3. During training, the objective is to maximize the log-likelihood of final states given by the reverse diffusion process, with final states sampled from the target state distribution $q(s_T|g)$,

$$
\begin{aligned}
L(\theta) &= \mathbb{E}_{s_T \sim q(s_T)} \left[ \log p_\theta(s_T) \right] = \int ds_T \, q(s_T) \log p_\theta(s_T) \\
&= \int ds_T \, q(s_T) \log \left[ \int ds_1 \ldots ds_{T-1} \, q(s_1, \ldots, s_{T-1}|s_T) p(s_1) \prod_{t=1}^{T-1} \frac{p_\theta(s_{t+1}|s_t)}{q(s_t|s_{t+1})} \right] \\
&\geq \int ds_1 \ldots ds_T \, q(s_1, \ldots, s_T) \log \left[ p(s_1) \prod_{t=1}^{T-1} \frac{p_\theta(s_{t+1}|s_t)}{q(s_t|s_{t+1})} \right]
\end{aligned}
$$

   where the lower bound is provided by Jensen's inequality.

We separate the term corresponding the initial state $s_1$,

$$L(\theta) \geq \int ds_1 \ldots ds_T \ q(s_1, \ldots, s_T) \sum_{t=1}^{T-1} \log \left[ \frac{p_\theta(s_{t+1}|s_t)}{q(s_t|s_{t+1})} \right] + \int ds_1 \ q(s_1) \log p(s_1)$$

$$= \sum_{t=1}^{T-1} \int ds_t ds_{t+1} \ q(s_t, s_{t+1}) \log \left[ \frac{p_\theta(s_{t+1}|s_t)}{q(s_t|s_{t+1})} \right] + \int ds_1 \ q(s_1) \log p(s_1)$$

We apply Bayes' rule to rewrite in terms of posterior of the forward diffusion,

$$L(\theta) \geq \sum_{t=1}^{T-1} \int ds_t ds_{t+1} \ q(s_t, s_{t+1}) \log \left[ \frac{p_\theta(s_{t+1}|s_t)}{q(s_{t+1}|s_t)} \frac{q(s_{t+1})}{q(s_t)} \right] + \int ds_1 \ q(s_1) \log p(s_1)$$

$$= \sum_{t=1}^{T-1} \int ds_t ds_{t+1} \ q(s_t, s_{t+1}) \log \left[ \frac{p_\theta(s_{t+1}|s_t)}{q(s_{t+1}|s_t)} \right]$$

$$+ \sum_{t=1}^{T-1} [H_q(S_t) - H_q(S_{t+1})] + \int ds_1 \ q(s_1) \log p(s_1)$$

$$+ H_q(S_1) - H_q(S_T) + \int ds_1 \ q(s_1) \log p(s_1)$$

$$= - \sum_{t=1}^{T-1} \int ds_t ds_{t+1} \ q(s_t) q(s_{t+1}|s_t) \log \left[ \frac{q(s_{t+1}|s_t)}{p_\theta(s_{t+1}|s_t)} \right]$$

$$+ H_q(S_1) - H_q(S_T) + \int ds_1 \ q(s_1) \log p(s_1)$$

$$= - \sum_{t=1}^{T-1} \int ds_t \ q(s_t) D_{KL} \big( q(s_{t+1}|s_t) \| p_\theta(s_{t+1}|s_t) \big)$$

$$+ H_q(S_1) - H_q(S_T) + \int ds_1 \ q(s_1) \log p(s_1)$$

4. We maximize the log-likelihood with respect to the policy parameters $\theta$, which is equivalent to minimizing the first term,

$$\theta^* = \arg\max_\theta L(\theta) \equiv \arg\min_\theta \sum_{t=1}^{T-1} \int ds_t \ q(s_t) D_{KL} \big( q(s_{t+1}|s_t) \| p_\theta(s_{t+1}|s_t) \big)$$

The posterior of the forward diffusion is simply the state transition using the behavior policy $\pi_\beta$,

$$\theta^* = \arg\min_\theta \sum_{t=1}^{T-1} \int ds_t \ q(s_t) D_{KL} \big( \mathcal{P}(s_{t+1}|s_t, \pi_\beta(\cdot)) \| \mathcal{P}(s_{t+1}|s_t, \pi_\theta(\cdot|s_t)) \big)$$

(a) For deterministic state transitions, the next state $s_{t+1}$ is given by the dynamics function $f$ of the MDP, $s_{t+1} = f(s_t, a_t)$. For a given state $s_t$, this dynamics function represents a fixed parameter transformation of the policy function. We exploit the property that KL divergence is invariant under parameter transformations. Thus for a deterministic MDP,

$$\theta^* = \arg\min_\theta \sum_{t=1}^{T-1} \int ds_t \ q(s_t) D_{KL} \big( f(s_t, \pi_\beta(\cdot)) \| f(s_t, \pi_\theta(\cdot|s_t)) \big)$$

$$= \arg\min_\theta \sum_{t=1}^{T-1} \int ds_t \ q(s_t) D_{KL} \big( \pi_\beta(\cdot) \| \pi_\theta(\cdot|s_t) \big)$$

(b) For stochastic state transitions, the next state $s_{t+1}$ is given by a noisy dynamics function $s_{t+1} = f(s_t, a_t, \epsilon)$, where $\epsilon \sim \xi(\epsilon)$ denotes random noise to account for the stochasticity.

For a given state $s_t$, this dynamics function represents a fixed parameter transformation of the joint distribution of the policy and the noise distribution. Abusing notation, we denote this joint distribution as $p(\pi, \xi)$. Since KL divergence is invariant under parameter transformations, for a stochastic MDP,

$$\theta^* = \arg\min_\theta \sum_{t=1}^{T-1} \int ds_t \ q(s_t) D_{KL}\big(f(s_t, \pi_\beta(\cdot), \xi) \| f(s_t, \pi_\theta(\cdot|s_t), \xi)\big)$$

$$= \arg\min_\theta \sum_{t=1}^{T-1} \int ds_t \ q(s_t) D_{KL}\big(p(\pi_\beta(\cdot), \xi) \| p(\pi_\theta(\cdot|s_t), \xi)\big)$$

Since the policy and the noise distribution $\xi$ are independent, the KL divergence decomposes,

$$\theta^* = \arg\min_\theta \sum_{t=1}^{T-1} \int ds_t \ q(s_t) D_{KL}\big(\pi_\beta(\cdot) \| \pi_\theta(\cdot|s_t)\big) + \sum_{t=1}^{T-1} D_{KL}\big(\xi \| \xi\big)$$

$$= \arg\min_\theta \sum_{t=1}^{T-1} \int ds_t \ q(s_t) D_{KL}\big(\pi_\beta(\cdot) \| \pi_\theta(\cdot|s_t)\big)$$

Minimizing the KL divergence between the policies is equivalent to maximizing the log-likelihood of the behavior policy action under the parameterized policy. Therefore, given state-action-goal tuples $(s, a) \sim \mathcal{D}(g)$,

$$\theta^* = \arg\max_\theta \mathbb{E}_{(s,a)\sim\mathcal{D}(g)} \big[\log \pi_\theta(a|s)\big]$$

Therefore, behavior cloning is equivalent to maximizing a lower bound on the log-likelihood of the target final states achieved by the reverse diffusion process.

## B ADDITIONAL ILLUSTRATIVE EXPERIMENTS

### B.1 MULTIPLE GOAL SETTING

The illustrative example presented in Section 4 considered a single goal setting. In this section, we verify that Merlin works as expected in the multiple goal setting. In these experiments, the forward diffusion process comprises taking random actions starting from one of the goals, which is picked randomly. A trained agent should be able to effectively navigate towards any one of these goals.

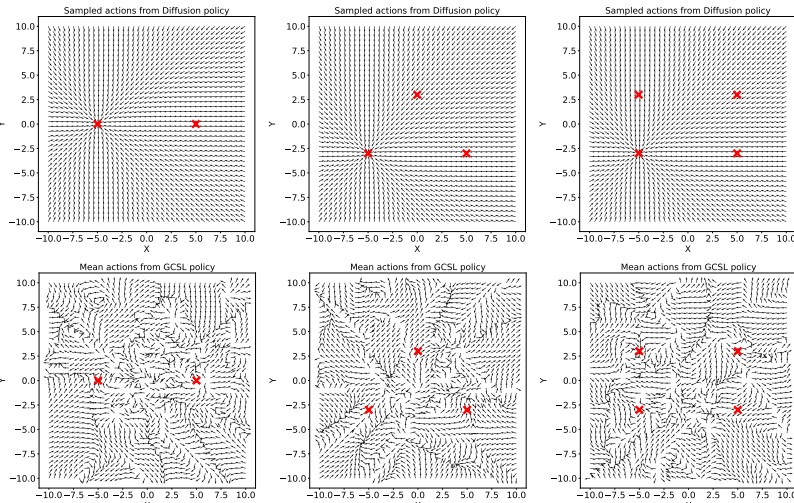

Figure 7: The policy is conditioned on the bottom leftmost goal in all cases. **Top:** Diffusion; **Bottom:** GCSL.

Figure 7 shows the predicted actions from Merlin and GCSL for navigating towards one particular goal (fixed to be the bottom leftmost goal) among two, three, and four possible goals. Merlin successfully navigates to the specified goal taking the most optimal path in all cases, whereas GCSL struggles to reach the specified goal.

## B.2 FOUR ROOMS NAVIGATION

The illustrative example presented in Section 4 considered a simple navigation problem. In this section, we extend the analysis to the four rooms variant, which adds walls that the agent must navigate around in order to reach the goal. We seek to understand whether Merlin can learn policies that produce more complex behavior compared to simply heading straight toward the goal.

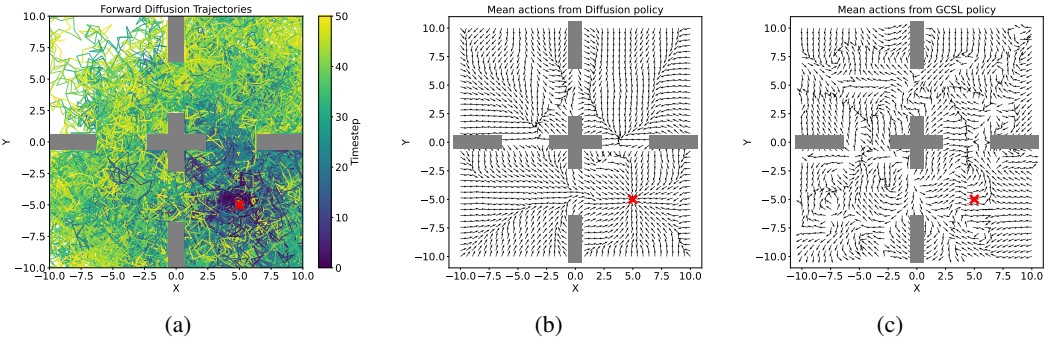

Figure 8: (a) Visualization of trajectories starting from the goal **X** generated during the forward process, (b) Predicted actions from policy trained via diffusion, (c) Predicted actions from policy trained using GCSL.

The goal state during training is fixed to $g = (5, -5)$ in one of the quadrants. Figure 8a visualizes the trajectories during forward diffusion by taking random actions starting from the goal. Figure 8b and Figure 8c visualize the policy learned by Merlin and GCSL, respectively. Both methods were trained for $100k$ policy updates. Merlin effectively learns to navigate around the walls, while still managing to reach the goal. In contrast, GCSL often navigates directly into the walls and in some areas wanders away from the goal.

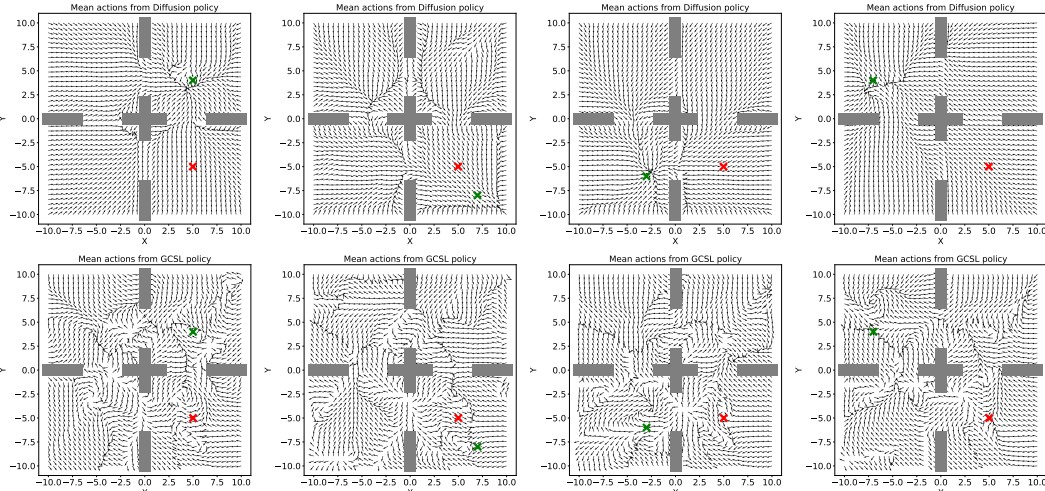

Figure 9: Evaluating the trained policy on out-of-distribution goals. Red **X** denotes the goal used during training, and green **X** denotes the goal used for evaluation. **Top:** Diffusion; **Bottom:** GCSL.

We then evaluate the trained policy on out-of-distribution goals. During training, the goal is fixed to $g = (5, -5)$ but during evaluation, we condition the policy on random goals. As shown in Figure 9, Merlin effectively generalizes this complex navigation behavior to new goals, learning to avoid the walls in most cases. One interesting case is when the goal is in the quadrant furthest away from the

training goal. Here, Merlin has some difficulty navigating around the walls, particularly the walls in the center. This is likely due to insufficient data generated by the forward diffusion process in this quadrant (see Figure 8a).

## C  RELATION TO DIFFUSION PROBABILISTIC MODELS

Merlin takes inspiration from generative diffusion models, resulting in several parallels as highlighted earlier. Notably, Figure 2 visualizes the noisy trajectories generated via the forward diffusion process and conveys the similarity of the learned policy to the score function in generative diffusion models. However, there are several key differences:

- The noise in diffusion probabilistic models is fixed to be Gaussian, whereas in application to RL, the noise corresponds to taking actions and reversing the dynamics, which is dependent on the properties of the MDP.
- The noisy samples for diffusion models lie outside the data manifold and hold no significance, while in this case, the noisy samples are valid states of the MDP.
- Lastly, conditioning the policy on goals is different from class conditioning in diffusion - here, any state in the diffusion path is a potential goal.

## D  IMPLEMENTATION DETAILS

### D.1  MERLIN ALGORITHM

---
**Algorithm 2** Reverse Model-based Rollout

---
**Input:** Dataset $\mathcal{D}$, number of new trajectories to collect $M$, number of training steps $N$
**Output:** Augmented dataset $\mathcal{D}_{\text{new}}$
$\mathcal{D}_{\text{new}} \leftarrow \mathcal{D}$
**for** $n \leftarrow 1, \ldots, N$ **do**
    Update $f_\psi$ by minimizing Equation (4)
    Update $\hat{D}_\xi$ by minimizing Equation (5)
**end for**
**for** $m \leftarrow 1, \ldots, M$ **do**
    Sample random final state $s_T$ from $\mathcal{D}$
    $\tau_{\text{new}} \leftarrow \{s_T\}$
    $s_{\text{current}} \leftarrow s_T$
    **for** $t \leftarrow T, \ldots, 1$ **do**
        Sample $z \sim \mathcal{N}(0, \mathbf{I})$
        $a_{\text{prev}} \leftarrow \hat{D}_\xi(s_{\text{current}}, z)$
        $s_{\text{prev}} \leftarrow f_\psi(s_{\text{current}}, a_{\text{prev}})$
        $\tau_{\text{new}} \leftarrow \{s_{\text{prev}}, a_{\text{prev}}\} \cup \tau_{\text{new}}$
        $s_{\text{current}} \leftarrow s_{\text{prev}}$
    **end for**
    $\mathcal{D}_{\text{new}} \leftarrow \mathcal{D}_{\text{new}} \cup \tau_{\text{new}}$
**end for**
**Return:** $\mathcal{D}_{\text{new}}$

---

---
**Algorithm 3** Merlin

---
**Input:** Dataset $\mathcal{D}$, hindsight ratio $p$, number of training steps $N$
**Output:** Policy $\pi_\theta$
Initialize policy $\pi_\theta$
$\mathcal{D}_{\text{new}} \leftarrow \text{ReverseModel}(\mathcal{D})$     ▷ For Merlin-P
$\mathcal{D}_{\text{new}} \leftarrow \text{TrajectoryStitch}(\mathcal{D})$ ▷ For Merlin-NP
**for** $n \leftarrow 1, \ldots, N$ **do**
    Sample batch $(s, a, g)$ from $\mathcal{D}_{\text{new}}$
    Relabel fraction $p$ of batch
    Update policy $\pi_\theta$ as per Equation (3)
    $\tau_{\text{new}} \leftarrow \{s_T\}$
**end for**
**Return:** $\pi_\theta$

---

### D.2  MERLIN: DETAILS OF POLICY NETWORK AND HYPERPARAMETERS

The policy is parameterized as a diagonal Gaussian distribution using an MLP with three hidden layers of 256 units each with the ReLU activation function, except for the final layer. The input to the policy comprises the state, the desired goal, and the time horizon. The time horizon is encoded using sinusoidal positional embeddings of 32 dimensions with the maximum period set to $T = 50$ since that is the maximum length of the trajectory for all our tasks. The output of the policy is the

mean and the standard deviation of the action. The $\tanh(\cdot)$ function is applied to the mean and it is multiplied by the maximum value of the action space to ensure the mean is within the correct range. The softplus function, $\text{softplus}(x) = \log(1 + \exp(x))$ is applied to the standard deviation to ensure non-negativity.

The policy was trained for $500k$ mini-batch updates using Adam optimizer with a learning rate of $5 \times 10^{-4}$ and a batch size of 512. The same policy network architecture and corresponding hyperparameters are used for all variations of Merlin. Merlin involves two main hyperparameters - the hindsight ratio and the time horizon used during evaluation. We perform ablations in Section 6.1 and report the tuned values for each task below.

Table 2: Optimal values for the hindsight ratio and time horizon for Merlin.

| Task Name | Hindsight Ratio | | Time Horizon | |
| --- | --- | --- | --- | --- |
| | Expert | Random | Expert | Random |
| PointReach | 0.2 | 1.0 | 1 | 1 |
| PointRooms | 0.2 | 1.0 | 1 | 1 |
| Reacher | 0.2 | 1.0 | 5 | 5 |
| SawyerReach | 0.2 | 1.0 | 1 | 1 |
| SawyerDoor | 0.2 | 1.0 | 5 | 5 |
| FetchReach | 0.2 | 1.0 | 1 | 1 |
| FetchPush | 0.0 | 0.2 | 20 | 20 |
| FetchPick | 0.0 | 0.5 | 10 | 50 |
| FetchSlide | 0.2 | 0.8 | 10 | 10 |
| HandReach | 1.0 | 1.0 | 1 | 1 |

### D.3 MERLIN-P: DETAILS OF REVERSE DYNAMICS MODEL AND REVERSE POLICY

Merlin-P uses a learned parametric reverse dynamics model and a reverse policy to simulate the forward diffusion process starting from potential goal states. We follow the procedure described in Wang et al. (2021), which is summarized here.

The reverse dynamics model $\hat{\mathcal{P}}_\psi(s|s', a)$ produces the previous state given the next state and a candidate action. Unlike Wang et al. (2021), we do not learn the reward model since Merlin does not require rewards for the learning process. The model parameters are optimized by minimizing the negative log-likelihood, which is equivalent to the mean squared error for deterministic environments,

$$\mathcal{L}_1(\psi) = \mathbb{E}_{(s,a,s') \sim \mathcal{D}} \left[ -\log \hat{\mathcal{P}}_\psi(s|s', a) \right] = \mathbb{E}_{(s,a,s') \sim \mathcal{D}} \|s - f_\psi(s', a)\|_2^2, \tag{4}$$

where $f_\psi(\cdot, \cdot)$ denotes the deterministic reverse dynamics function. The network architecture is an MLP with three hidden layers of 256 units each with the ReLU activation function. The dynamics model is trained for 20 epochs using Adam optimizer with a learning rate of $3 \times 10^{-4}$ and a batch size of 256.

To generate diverse candidate actions for reverse rollouts, the reverse policy is parameterized as a conditional variational autoencoder (CVAE), consisting of an action encoder $\hat{E}_\omega(s', a)$ that outputs a latent vector $z$, and an action decoder $\hat{D}_\xi(s', z)$ which reconstructs the action given latent vector $z$. The reverse policy is trained by maximizing the variational lower bound,

$$\mathcal{L}_2(\omega, \xi) = \mathbb{E}_{(s,a,s') \sim \mathcal{D}, z \sim \hat{E}_\omega(s',a)} \left[ \left( a - \hat{D}_\xi(s', z) \right)^2 + D_{KL} \left( \hat{E}_\omega(s', a) \| \mathcal{N}(0, \mathbf{I}) \right) \right] \tag{5}$$

The encoder is an MLP with two hidden layers of 256 units each using the ReLU activation function. The latent space dimension is twice the action space dimension. The encoder outputs the mean and log standard deviation, the latter is clamped to $[-4, 15]$ for numerical stability. The decoder is also an MLP with two hidden layers of 256 units each using the ReLU activation function. The $\tanh(\cdot)$ function is applied to the action output of the decoder and it is multiplied by the maximum value of the action space to ensure it is within the correct range. The CVAE is trained for 20 epochs using Adam optimizer with a learning rate of $3 \times 10^{-4}$ and a batch size of 256.

In order to generate a rollout starting from state $s'$, a latent vector is drawn from the standard Gaussian distribution, $\tilde{z} \sim \mathcal{N}(0, \mathbf{I})$. The action decoder is used to obtain a candidate action $\tilde{a} = \hat{D}_\xi(s', \tilde{z})$, and finally the reverse dynamics model produces the previous state $\tilde{s} = f_\psi(s', \tilde{a})$. As shown in Section 6, this method works well for simple environments with relatively low-dimension state spaces. In higher dimensions, the compounding model error produces unrealistic rollouts, especially over long horizons.

### D.4 MERLIN-NP: DETAILS OF NEAREST-NEIGHBOR TRAJECTORY STITCHING

We propose a novel trajectory stitching method in Section 5.1 which is used for Merlin-NP. The method is based on finding the nearest neighbor of states along a trajectory, which involves choosing a metric. We chose the Euclidean ($\ell_2$) distance due to its wide use in practice, but other choices for metrics such as the Manhattan distance ($\ell_1$) would also be suitable. The use of distance metrics in high dimensions can be unreliable, however, note that all methods implicitly assume a metric. The policy and the value function assume a metric to determine which states are similar, and therefore, should yield similar actions or values respectively.

In order to search for nearest neighbors as efficiently as possible, we construct a ball tree from all the states in the dataset, instead of a KD tree. The ball tree partitions the space using a series of hyperspheres instead of partitioning along the Cartesian axes, which leads to more efficient queries in higher dimensions. The query time for the ball tree grows as approximately $O(d \log N)$ for $N$ samples of $d$-dimensional data. For a KD tree, the query time is the same as a ball tree for lower dimensions ($< 20$), however, it quickly becomes comparable to a brute force search for higher dimensions.

Table 3: Values of distance threshold $\delta$ used for Merlin-NP.

| Task Name | $\delta$ |
|---|---|
| PointReach, PointRooms | $1 \times 10^{-6}$ |
| Reacher | $1 \times 10^{-1}$ |
| SawyerReach, SawyerDoor | $1 \times 10^{-5}$ |
| FetchReach, FetchPush | $5 \times 10^{-3}$ |
| FetchPick, FetchSlide | $1 \times 10^{-2}$ |
| HandReach | $2$ |

For the trajectory stitching method, we also choose a distance threshold $\delta$ which determines which states are considered similar enough to allow stitching. Very large values of $\delta$ would result in constant switching between trajectories even when the states are considerably dissimilar, leading to mismatched state-action pairs at the stitching point. On the other hand, if the value of $\delta$ is too small, then no states would be considered similar enough, leaving us with the original dataset. Since the properties of the state space are different for different tasks, the threshold value has to be tuned separately for each scenario. We chose the value such that on average, there would be one or two stitching operations per trajectory. Table 3 presents the value of distance threshold $\delta$ for each task.

We collect 2000 stitched trajectories for the simpler tasks (PointReach, PointRooms, Reacher, SawyerReach, SawyerDoor and FetchReach), which effectively doubles the amount of offline data. For the harder tasks (FetchPush, FetchPick, FetchSlide and HandReach), we collect 10000 stitched trajectories to augment the 40000 trajectories in the original dataset.

### D.5 COMPUTE

Figure 10 shows the training and inference times averaged over all tasks for each method in Section 6. The diffusion-based baselines (DD and g-DQL) have significantly higher training and inference times since each environment step requires denoising the entire reverse diffusion chain. In contrast, Merlin has comparable training and inference times to non-diffusion-based offline GCRL methods, which is roughly an order of magnitude lower. Merlin-P suffers from an overhead compared to Merlin due to training the reverse dynamics model and the reverse policy. The training time overhead for Merlin-NP is during the trajectory stitching phase for nearest-neighbors search.

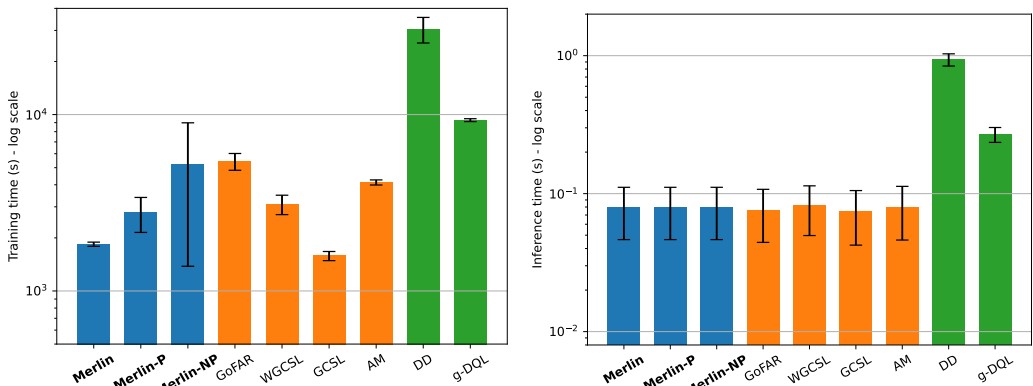

Figure 10: Mean training and inference times over different tasks for each method. Training times are reported for $500k$ policy updates and inference times are reported for one episode comprising of 50 time steps.

The large variation in training time for Merlin-NP is because trajectory stitching for the harder tasks (FetchPush, FetchPick, FetchSlide and HandReach) takes more time owing to the higher state-space dimension and a larger number of collected trajectories.

## E  BASELINE IMPLEMENTATION DETAILS

### E.1  OFFLINE GCRL METHODS

For all the methods described in this section, the policy architecture is identical to the one used for Merlin, described in Appendix D. Wherever applicable, the critic architecture is an MLP with three hidden layers of 256 units each with the ReLU activation. All of these methods were fine-tuned in Ma et al. (2022), and we used their implementation (https://github.com/JasonMa2016/GoFAR/tree/main) to produce the baseline results in Section 6.

**GCSL.**  GCSL uses hindsight relabeling by setting the goal to be a future state within the same trajectory, where future states are sampled uniformly from possible choices. The policy is learned using behavior cloning,

$$\max_{\pi} \mathbb{E}_{(s,a,g)\sim\mathcal{D}_{\text{relabel}}} \left[\log \pi(a|s,g)\right]$$

**WGCSL.**  This method builds upon GCSL and learns a Q-function with standard TD learning, where the dataset $\mathcal{D}$ uses hindsight relabeling and $\bar{Q}$ denotes the stop-gradient operation,

$$\min_{Q} \mathbb{E}_{(s_t,a_t,s_{t+1},g)\sim\mathcal{D}} \left[\left(r(s_t,g) + \gamma\bar{Q}(s_{t+1},\pi(s_{t+1},g),g) - Q(s_t,a_t,g)\right)^2\right]$$

The advantage function is defined as $A(s_t,a_t,g) = r(s_t,g) + \gamma Q(s_{t+1},\pi(s_{t+1},g),g) - Q(s_t,\pi(s_t,g),g)$, and is used to weight the regression loss for policy updates,

$$\max \pi \mathbb{E}_{(s_t,a_t,\phi(s_i))\sim\mathcal{D}} \left[\gamma^{i-t} \exp_{\text{clip}}(A(s_t,a_t,\phi(s_i))) \log \pi(a_t|s_t,\phi(s_i))\right]$$

**Actionable Model.**  AM employs an actor-critic framework similar to DDPG (Lillicrap et al., 2015), but uses conservative critic updates by adding a regularization term to the regular TD updates,

$$\min_{Q} \mathbb{E}_{(s_t,a_t,s_{t+1},g)\sim\mathcal{D}} \left[\left(r(s_t,g) + \gamma\bar{Q}(s_{t+1},\pi(s_{t+1},g),g) - Q(s_t,a_t,g)\right)^2 + \mathbb{E}_{a\sim\exp(\bar{Q})}[Q(s,a,g)]\right]$$

The policy updates are similar to DDPG, where gradients are backpropagated through the critic,

$$\max_{\pi} \mathbb{E}_{(s_t,a_t,s_{t+1},g)\sim\mathcal{D}} \left[Q(s_t,\pi(s_t,g),g)\right]$$

In addition to hindsight relabeling, AM uses a goal-chaining technique where for half of the relabeled transitions in each minibatch, the relabelled goals are randomly sampled from the offline dataset.

**GoFAR.** GoFAR takes a state-occupancy matching perspective by training a discriminator to define a reward function that encourages visiting states that occur more often in conjunction with the desired goal,

$$\min_c \mathbb{E}_{g \sim p(g)} \left[ \mathbb{E}_{p(s,g)} \left[ \log c(s,g) \right] + \mathbb{E}_{(s,g) \sim \mathcal{D}} \left[ \log(1 - c(s,g)) \right] \right]$$

where $p(s,g) = \exp(r(s,g))/Z$ and $Z = \int \exp(r(s,g))$. The reward function used for learning the critic is $R(s,g) = -\log\left(1/c(s,g) - 1\right)$. GoFAR also uses $f$-divergence regularization to learn a value function,

$$\min_{V(s,g) \geq 0} (1 - \gamma) \mathbb{E}_{s \sim \mu(s), g \sim p(g)}[V(s,g)] + \mathbb{E}_{(s,a,g) \sim \mathcal{D}}[f_\star(R(s,g) + \gamma \mathbb{E}_{s' \sim \mathcal{P}(\cdot|s,a)}[V(s',g)] - V(s,g))]$$

where $f_\star$ denotes the convex conjugate of $f$. The policy is updated using regression weights that are first-order derivatives of $f_\star$ evaluated at the optimal advantage,

$$\max_\pi \mathbb{E}_{g \sim p(g)} \mathbb{E}_{(s,a) \sim \mathcal{D}} \left[ f'_\star(R(s,g) + \gamma \mathbb{E}_{s' \sim \mathcal{P}(\cdot|s,a)}[V^*(s',g)] - V^*(s,g)) \log \pi(a|s,g) \right]$$

where $V^*$ denotes the optimal value function obtained after training. GoFAR does not use hindsight relabeling.

## E.2 DIFFUSION-BASED METHODS

**Decision Diffuser.** Decision diffuser models sequential decision-making as a conditional generative modeling problem,

$$\max_\theta E_{\tau \sim \mathcal{D}}[\log p_\theta(\mathbf{x}_0(\tau)|\mathbf{y}(\tau))]$$

where $\mathbf{y}(\tau)$ denotes the conditioning variable representing returns, goals, or other constraints that are desirable in the generated trajectory. The forward and reverse diffusion process are $q(\mathbf{x}_{k+1}(\tau)|\mathbf{x}_k(\tau))$ and $p_\theta(\mathbf{x}_{k-1}(\tau)|\mathbf{x}_k(\tau), \mathbf{y}(\tau))$. The diffusion is performed on state sequences, and actions are obtained using an inverse dynamics model $a_t = f_\psi(s_t, s_{t+1})$. The reverse diffusion process learns a conditional denoising function by sampling noise $\epsilon \sim \mathcal{N}(\mathbf{0}, \mathbf{I})$ and a timestep $k \sim \mathcal{U}\{1, \dots, K\}$,

$$\min_{\theta, \psi} \mathbb{E}_{k, \tau \in D, \beta \sim \text{Bern}(p)} \left[ \|\epsilon - \epsilon_\theta \left(\mathbf{x}_k(\tau), (1-\beta)\mathbf{y}(\tau) + \beta\varnothing, k\right)\|_2^2 \right] + \mathbb{E}_{(s,a,s') \in \mathcal{D}} \left[ \|a - f_\psi(s,s')\|_2^2 \right]$$

Classifier-free guidance is employed during planning to generate trajectories respecting the conditioning variable $\mathbf{y}(\tau)$ by starting with Gaussian noise $\mathbf{x}_K(\tau)$ and refining $\mathbf{x}_k(\tau)$ into $\mathbf{x}_{k-1}(\tau)$ at each intermediate timestep with the perturbed noise,

$$\epsilon = \epsilon_\theta \left(\mathbf{x}_k(\tau), \varnothing, k\right) + \omega \left(\epsilon_\theta \left(\mathbf{x}_k(\tau), \mathbf{y}(\tau), k\right) - \epsilon_\theta \left(\mathbf{x}_k(\tau), \varnothing, k\right)\right)$$

The denoising function is a temporal U-Net model with residual blocks. We used the official implementation provided here: https://github.com/anuragajay/decision-diffuser/tree/main.

**Diffusion QL.** The policy is represented via the reverse process of a conditional diffusion model, where the end sample of the reverse chain, $a^0$, is the action used for RL evaluation,

$$\pi_\theta(a|s) = p_\theta(a^{0:N}|s) = \mathcal{N}(a^N; \mathbf{0}, \mathbf{I}) \prod_{i=1}^N p_\theta(a^{i-1}|a^i, s)$$

The reverse process is modeled as a noise prediction model with fixed variance $\Sigma_i = \beta_i \mathbf{I}$. The mean is reparameterized in terms of a learned denoising function $\epsilon_\theta$, which is trained using the simplified objective proposed in Ho et al. (2020),

$$\min_\theta \mathbb{E}_{i \sim U\{1, \dots, N\}, \epsilon \sim \mathcal{N}(\mathbf{0}, \mathbf{I}), (s,a) \sim \mathcal{D}} \left[ \|\epsilon - \epsilon_\theta \left(\sqrt{\bar{\alpha}_i} a + \sqrt{1 - \bar{\alpha}_i}\epsilon, s, i\right)\|_2^2 \right]$$

To sample actions from the policy, first sample $a^N \sim \mathcal{N}(\mathbf{0}, \mathbf{I})$ and denoise using the denoising model for $N$ steps,

$$a^{i-1} \mid a^i = \frac{a^i}{\sqrt{\alpha_i}} - \frac{\beta_i}{\sqrt{\alpha_i(1 - \bar{\alpha}_i)}}\epsilon_\theta(a^i, s, i) + \sqrt{\beta_i}\epsilon, \quad \epsilon \sim \mathcal{N}(\mathbf{0}, \mathbf{I}), \text{ for } i = N, \dots, 1.$$

Similar to DDPG, Diffusion-QL also uses a learned critic function trained using the standard TD error and backpropagates through the critic during training to prefer actions with high Q-values. In order to apply this method to goal-conditioned tasks, we additionally condition the policy on the goal. The architecture of the policy is a three-layer MLP with 256 hidden units each and the Mish activation function. The critic similarly has three layers of 256 units each and Mish activations. We used the official implementation provided here: https://github.com/Zhendong-Wang/Diffusion-Policies-for-Offline-RL.

## F    TASK DESCRIPTIONS

We consider 10 different goal-conditioned tasks with sparse and binary rewards. The state, action, and goal spaces are continuous, and the maximum length of each episode is set as 50. We use the offline benchmark introduced in Yang et al. (2021). The relatively easier tasks (PointReach, PointRooms, Reacher, SawyerReach, SawyerDoor, and FetchReach) have 2000 trajectories each ($1 \times 10^5$ transitions); and the harder tasks (FetchPush, FetchPick, FetchSlide, and HandReach) have 40000 trajectories ($2 \times 10^6$ transitions).

The benchmark consists of two settings 'expert' and 'random'. The 'expert' dataset consists of trajectories collected by a policy trained using online DDQPG+HER with added Gaussian noise ($\sigma = 0.2$) to increase diversity, while the 'random' dataset consists of trajectories collected by sampling random actions.

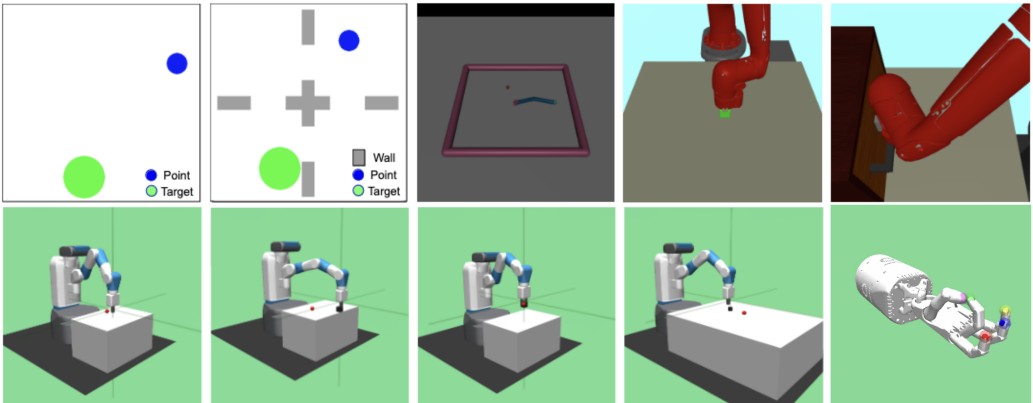

Figure 11: Goal-conditioned tasks from left to right, top to bottom: PointReach, PointRooms, Reacher, SawyerReach, SawyerDoor, FetchReach, FetchPush, FetchPick, FetchSlide, and HandReach.

**PointReach.**    The environment is adapted from *multiworld*[2]. The blue point represents the agent which is tasked with reaching the green circle representing the goal. The state space is two dimensional representing the $(x, y)$ coordinates of the blue point, where $(x, y) \in [-5, 5] \times [-5, 5]$. The actions space is also two dimensional representing the displacement in $x$ and $y$ directions, $a \in [-1, 1] \times [-1, 1]$. The goal space is the same as the state space, $\phi(s) = s$. The initial position of the agent and the goal are randomly initialized. Success is defined if the agent reaches within a certain radium of the goal. The reward function is defined as,

$$r(s_{XY}, a, g_{XY}) = \mathbb{1}[\|s_{XY} - g_{XY}\|_2^2 \le \epsilon]$$

where the tolerance is $\epsilon = 1$.

**PointRooms.**    The environment is a variation of PointReach environment. The task is again for the blue dot representing the agent to reach the green circle, however, there are vertical and horizontal walls forming four rooms, which make navigation more challenging. The reward function, and the state, action, and goal spaces are the same as in PointReach.

---

[2]https://github.com/vitchyr/multiworld

**Reacher.**   The environment is included in Gymnasium[3]. Reacher is a two-jointed robot arm tasked with moving the robot's end effector close to a target that is spawned at a random position. The state space is 11-dimensional representing the angles, positions and velocities of the joints. The goals are $(x, y)$ coordinates of the target, and $\phi(s) = s[4:6]$. The two-dimensional actions represent the torque applied at each joint. The reward function is defined as,

$$r(s_{XY}, a, g_{XY}) = \mathbb{1}[\|s_{XY} - g_{XY}\|_2^2 \leq \epsilon]$$

where the tolerance is $\epsilon = 0.05$.

**SawyerReach.**   The environment is adapted from *multiworld*. The Sawyer robot is tasked with reaching a target position using its end effector. The state space is 3-dimensional representing the $(x, y, z)$ coordinates of the end effector, and the goal space is also 3-dimensional representing the $(x, y, z)$ coordinates of the target position, $\phi(s) = s$. The 3-dimensional actions describe the coordinates of the next position of the end effector. The reward function is defined as,

$$r(s_{XYZ}, a, g_{XYZ}) = \mathbb{1}[\|s_{XYZ} - g_{XYZ}\|_2^2 \leq \epsilon]$$

where the tolerance is $\epsilon = 0.06$.

**SawyerDoor.**   The environment is adapted from *multiworld*. The Sawyer robot is tasked with opening a door to a specified angle. The 4-dimensional state space represents the coordinates of the end effector of the robot and the angle of the door. The action space is the 3-dimensional next position of the end effector. The goal is the desired angle of the door, $\phi(s) = s[-1]$, which is between $[0, 0.83]$ radians. The reward function is defined as,

$$r(s, a, g) = \mathbb{1}[|\phi(s) - g| \leq \epsilon]$$

where the tolerance is $\epsilon = 0.06$.

**FetchReach.**   The environment is included in Gymnasium-Robotics[4]. It consists of a 7-DoF robotic arm, with a two-fingered parallel gripper attached to it. The task is to reach a target location which is specified as a 3-dimensional goal representing the $(x, y, z)$ coordinates of the target location. The states are 10-dimensional represent the kinematic information of the end effector, including the positions and velocities of the end effector and the gripper joint displacement. The actions are 4-dimensional which describes the displacement of the end effector, and the last dimension which represents the gripper opening/closing is not used for this task. The state-to-goal mapping is $\phi(s) = s[0:3]$. The reward function is defined as,

$$r(s, a, g_{XYZ}) = \mathbb{1}[\|\phi(s) - g_{XYZ}\|_2^2 \leq \epsilon]$$

where the tolerance is $\epsilon = 0.05$.

**FetchPush.**   The environment is included in Gymnasium-Robotics. The 7-DoF robotic arm from FetchReach is tasked with pushing a block to a target location. The state space is 25- dimensional, including the gripper's position, linear velocities, and the box's position, rotation, linear and angular velocities. The 4-dimensional state space describes the displacement of the end effector and the gripper opening/closing. The goal is defined as the target $(x, y, z)$ position of the block, and the mapping is $\phi(s) = s[3:6]$. In this task, the block is always on top of the table, hence the block $z$ coordinate is always fixed. The reward function is defined as,

$$r(s, a, g_{XYZ}) = \mathbb{1}[\|\phi(s) - g_{XYZ}\|_2^2 \leq \epsilon]$$

where the tolerance is $\epsilon = 0.05$.

**FetchPick.**   The environment is included in Gymnasium-Robotics. The 7-DoF robotic arm from FetchReach is tasked with picking up a block and taking it to a target location specified as $(x, y, z)$ coordinates. The target $z$ coordinate of the block is not fixed and may be in the air above the table, requiring the robotic arm to pick up the block using the gripper. The state space, action space, goal space, state-to-goal mapping, and reward function are the same as FetchPush.

---

[3]https://github.com/Farama-Foundation/Gymnasium
[4]https://github.com/Farama-Foundation/Gymnasium-Robotics

**FetchSlide.** The environment is included in Gymnasium-Robotics. The 7-DoF robotic arm from FetchReach is tasked with moving a block to a target position specified as $(x, y, z)$ coordinates. The block is always on top of the table, hence the $z$ coordinate of the block is always fixed. However, the $(x, y)$ coordinates of the target position are out of reach of the robotic arm, hence it must hit the block with the appropriate amount of force for it to slide and then stop at the goal position. The state space, action space, goal space, state-to-goal mapping, and reward function are the same as FetchPush.

**HandReach.** The environment is included in Gymnasium-Robotics. A 24-DoF anthropomorphic hand is tasked with manipulating its fingers to reach a target configuration. The state space is 63-dimensional comprising two 24-dimensional vectors describing the positions and velocities of the joints, and five 3-dimensional vectors describing the $(x, y, z)$ positions of each fingertip. The 20-dimensional actions describe the absolute angular positions of the actuated joints. The goals are specified as 15-dimensional vectors, describing the $(x, y, z)$ coordinates of the fingertips with $\phi(s) = s[-15 :]$. The reward function is defined as,

$$r(s, a, g) = \mathbb{1}[\|\phi(s) - g\|_2^2 \le \epsilon]$$

where the tolerance is $\epsilon = 0.01$.

## G   GCSL WITH TRAJECTORY STITCHING

We apply a modified version of the nearest-neighbor trajectory stitching operation to GCSL and report the performance in Table 4 and Table 5, averaged over 10 seeds. The technique described in Section 5.1 applies to reverse trajectories, for GCSL we construct forward trajectories by adding the state-action pair succeeding the neighbors. We observe that this technique improves performance for most tasks, demonstrating it as a general-purpose data augmentation technique for offline GCRL.

Table 4: Discounted returns.

| | Task Name | Merlin | Merlin-NP | GCSL | GCSL+TS |
|---|---|---|---|---|---|
| **Expert** | PointReach | $29.26_{\pm0.04}$ | $29.30_{\pm0.05}$ | $22.85_{\pm1.26}$ | $23.22_{\pm1.71}$ |
| | PointRooms | $25.38_{\pm0.37}$ | $25.42_{\pm0.32}$ | $18.28_{\pm2.29}$ | $19.87_{\pm1.55}$ |
| | Reacher | $22.75_{\pm0.59}$ | $24.97_{\pm0.54}$ | $20.05_{\pm1.37}$ | $22.12_{\pm1.16}$ |
| | SawyerReach | $26.89_{\pm0.07}$ | $27.35_{\pm0.06}$ | $19.20_{\pm1.79}$ | $20.88_{\pm1.60}$ |
| | SawyerDoor | $26.18_{\pm2.19}$ | $26.15_{\pm2.08}$ | $20.12_{\pm1.33}$ | $20.61_{\pm1.26}$ |
| | FetchReach | $30.29_{\pm0.03}$ | $30.42_{\pm0.04}$ | $23.68_{\pm1.07}$ | $23.59_{\pm1.32}$ |
| | FetchPush | $19.91_{\pm1.20}$ | $21.58_{\pm1.63}$ | $17.58_{\pm1.47}$ | $19.15_{\pm1.29}$ |
| | FetchPick | $19.66_{\pm0.78}$ | $20.41_{\pm0.92}$ | $12.95_{\pm1.90}$ | $13.85_{\pm1.66}$ |
| | FetchSlide | $4.19_{\pm1.89}$ | $4.95_{\pm2.02}$ | $1.67_{\pm1.41}$ | $2.11_{\pm1.46}$ |
| | HandReach | $22.11_{\pm0.55}$ | $24.93_{\pm0.49}$ | $0.15_{\pm0.11}$ | $0.16_{\pm0.13}$ |
| **Random** | PointReach | $29.26_{\pm0.04}$ | $29.31_{\pm0.04}$ | $17.74_{\pm1.84}$ | $20.01_{\pm1.63}$ |
| | PointRooms | $24.80_{\pm0.36}$ | $25.16_{\pm0.59}$ | $14.69_{\pm2.51}$ | $16.05_{\pm1.97}$ |
| | Reacher | $21.09_{\pm0.65}$ | $22.24_{\pm0.54}$ | $10.62_{\pm2.30}$ | $12.89_{\pm2.34}$ |
| | SawyerReach | $26.70_{\pm0.14}$ | $26.86_{\pm0.07}$ | $8.78_{\pm2.59}$ | $9.12_{\pm2.26}$ |
| | SawyerDoor | $19.05_{\pm0.66}$ | $21.69_{\pm2.36}$ | $12.47_{\pm3.08}$ | $13.64_{\pm2.68}$ |
| | FetchReach | $30.42_{\pm0.04}$ | $30.42_{\pm0.04}$ | $18.96_{\pm1.77}$ | $19.58_{\pm1.72}$ |
| | FetchPush | $5.21_{\pm0.43}$ | $7.22_{\pm0.35}$ | $4.22_{\pm2.19}$ | $5.21_{\pm1.98}$ |
| | FetchPick | $3.75_{\pm0.18}$ | $4.36_{\pm0.19}$ | $0.81_{\pm0.82}$ | $0.95_{\pm0.90}$ |
| | FetchSlide | $2.67_{\pm0.35}$ | $3.15_{\pm0.14}$ | $0.24_{\pm0.27}$ | $0.31_{\pm0.36}$ |
| | HandReach | $14.89_{\pm2.54}$ | $17.61_{\pm3.06}$ | $1.41_{\pm0.51}$ | $2.06_{\pm0.76}$ |

Table 5: Success rates.

| | Task Name | Merlin | Merlin-NP | GCSL | GCSL+TS |
|---|---|---|---|---|---|
| **Expert** | PointReach | $1.00_{\pm0.00}$ | $1.00_{\pm0.00}$ | $1.00_{\pm0.00}$ | $1.00_{\pm0.00}$ |
| | PointRooms | $0.91_{\pm0.16}$ | $0.94_{\pm0.01}$ | $0.79_{\pm0.60}$ | $0.80_{\pm0.62}$ |
| | Reacher | $1.00_{\pm0.00}$ | $1.00_{\pm0.00}$ | $1.00_{\pm0.00}$ | $1.00_{\pm0.00}$ |
| | SawyerReach | $1.00_{\pm0.00}$ | $1.00_{\pm0.00}$ | $1.00_{\pm0.00}$ | $1.00_{\pm0.00}$ |
| | SawyerDoor | $0.95_{\pm0.08}$ | $0.94_{\pm0.11}$ | $0.84_{\pm0.16}$ | $0.85_{\pm0.14}$ |
| | FetchReach | $1.00_{\pm0.00}$ | $1.00_{\pm0.00}$ | $0.98_{\pm0.00}$ | $1.00_{\pm0.00}$ |
| | FetchPush | $0.95_{\pm0.05}$ | $0.96_{\pm0.04}$ | $0.88_{\pm0.09}$ | $0.89_{\pm0.10}$ |
| | FetchPick | $0.92_{\pm0.03}$ | $0.96_{\pm0.06}$ | $0.64_{\pm0.09}$ | $0.67_{\pm0.09}$ |
| | FetchSlide | $0.20_{\pm0.04}$ | $0.25_{\pm0.07}$ | $0.22_{\pm0.14}$ | $0.24_{\pm0.12}$ |
| | HandReach | $0.78_{\pm0.04}$ | $0.85_{\pm0.02}$ | $0.03_{\pm0.05}$ | $0.04_{\pm0.05}$ |
| **Random** | PointReach | $1.00_{\pm0.00}$ | $1.00_{\pm0.00}$ | $1.00_{\pm0.00}$ | $1.00_{\pm0.00}$ |
| | PointRooms | $0.89_{\pm0.02}$ | $0.92_{\pm0.02}$ | $0.77_{\pm0.11}$ | $0.79_{\pm0.12}$ |
| | Reacher | $0.98_{\pm0.02}$ | $1.00_{\pm0.00}$ | $0.80_{\pm0.06}$ | $0.84_{\pm0.07}$ |
| | SawyerReach | $1.00_{\pm0.00}$ | $1.00_{\pm0.00}$ | $0.91_{\pm0.09}$ | $0.93_{\pm0.11}$ |
| | SawyerDoor | $0.57_{\pm0.03}$ | $0.59_{\pm0.05}$ | $0.44_{\pm0.16}$ | $0.45_{\pm0.14}$ |
| | FetchReach | $1.00_{\pm0.00}$ | $1.00_{\pm0.00}$ | $0.96_{\pm0.05}$ | $0.98_{\pm0.03}$ |
| | FetchPush | $0.20_{\pm0.06}$ | $0.24_{\pm0.09}$ | $0.20_{\pm0.11}$ | $0.22_{\pm0.10}$ |
| | FetchPick | $0.12_{\pm0.01}$ | $0.18_{\pm0.01}$ | $0.06_{\pm0.08}$ | $0.07_{\pm0.06}$ |
| | FetchSlide | $0.11_{\pm0.02}$ | $0.20_{\pm0.04}$ | $0.06_{\pm0.08}$ | $0.06_{\pm0.07}$ |
| | HandReach | $0.49_{\pm0.05}$ | $0.62_{\pm0.07}$ | $0.04_{\pm0.04}$ | $0.04_{\pm0.04}$ |

# H  ADDITIONAL EXPERIMENTAL RESULTS

Table 6: Discounted returns, averaged over 10 seeds.

| | Task Name | Ours | | | Offline GCRL | | | | Diffusion-based | |
|---|---|---|---|---|---|---|---|---|---|---|
| | | Merlin | Merlin-P | Merlin-NP | GoFAR | WGCSL | GCSL | AM | DD | g-DQL |
| **Expert** | PointReach | $\mathbf{29.26}_{\pm0.04}$ | $29.17_{\pm0.15}$ | $\mathbf{29.30}_{\pm0.05}$ | $27.18_{\pm0.65}$ | $25.91_{\pm0.87}$ | $22.85_{\pm1.26}$ | $26.14_{\pm1.11}$ | $10.03_{\pm0.88}$ | $28.65_{\pm0.44}$ |
| | PointRooms | $25.38_{\pm0.37}$ | $25.25_{\pm0.07}$ | $\mathbf{25.42}_{\pm0.32}$ | $20.40_{\pm1.00}$ | $19.90_{\pm0.99}$ | $18.28_{\pm2.29}$ | $23.24_{\pm1.58}$ | $5.84_{\pm2.67}$ | $\mathbf{27.53}_{\pm0.57}$ |
| | Reacher | $22.75_{\pm0.59}$ | $23.25_{\pm0.17}$ | $\mathbf{24.97}_{\pm0.54}$ | $22.51_{\pm0.82}$ | $23.35_{\pm0.64}$ | $20.05_{\pm1.37}$ | $22.36_{\pm1.03}$ | $4.39_{\pm1.08}$ | $22.54_{\pm1.42}$ |
| | SawyerReach | $\mathbf{26.89}_{\pm0.07}$ | $25.05_{\pm0.60}$ | $\mathbf{27.35}_{\pm0.06}$ | $22.82_{\pm1.15}$ | $22.07_{\pm1.46}$ | $19.20_{\pm1.79}$ | $23.56_{\pm0.33}$ | $3.39_{\pm0.75}$ | $24.17_{\pm0.01}$ |
| | SawyerDoor | $\mathbf{26.18}_{\pm2.19}$ | $25.75_{\pm0.97}$ | $26.15_{\pm2.08}$ | $23.62_{\pm0.35}$ | $23.92_{\pm1.10}$ | $20.12_{\pm1.33}$ | $\mathbf{26.39}_{\pm0.42}$ | $7.85_{\pm0.77}$ | $24.81_{\pm0.38}$ |
| | FetchReach | $30.29_{\pm0.03}$ | $30.26_{\pm0.02}$ | $\mathbf{30.42}_{\pm0.04}$ | $29.21_{\pm0.26}$ | $28.17_{\pm0.38}$ | $23.68_{\pm1.07}$ | $29.08_{\pm0.12}$ | $1.55_{\pm0.68}$ | $28.71_{\pm0.15}$ |
| | FetchPush | $19.91_{\pm1.20}$ | $2.23_{\pm2.20}$ | $21.58_{\pm1.63}$ | $\mathbf{22.41}_{\pm1.69}$ | $\mathbf{22.22}_{\pm1.51}$ | $17.58_{\pm1.47}$ | $19.86_{\pm3.16}$ | $5.49_{\pm2.85}$ | $17.82_{\pm0.55}$ |
| | FetchPick | $19.66_{\pm0.78}$ | $1.43_{\pm1.01}$ | $\mathbf{20.41}_{\pm0.92}$ | $\mathbf{19.79}_{\pm1.12}$ | $18.32_{\pm1.56}$ | $12.95_{\pm1.90}$ | $17.04_{\pm3.81}$ | $2.76_{\pm0.64}$ | $14.45_{\pm0.61}$ |
| | FetchSlide | $4.19_{\pm1.89}$ | $0.00_{\pm0.00}$ | $\mathbf{4.95}_{\pm2.02}$ | $3.34_{\pm1.01}$ | $\mathbf{5.17}_{\pm3.17}$ | $1.67_{\pm1.41}$ | $3.31_{\pm1.46}$ | $1.21_{\pm0.59}$ | $0.98_{\pm0.59}$ |
| | HandReach | $22.11_{\pm0.55}$ | $0.00_{\pm0.00}$ | $\mathbf{24.93}_{\pm0.49}$ | $15.39_{\pm6.37}$ | $18.05_{\pm5.12}$ | $0.15_{\pm0.11}$ | $0.00_{\pm0.00}$ | $0.00_{\pm0.00}$ | $0.00_{\pm0.00}$ |
| | Average Rank | **2.7** | 5.3 | **1.6** | 4.5 | 4.6 | 7.3 | 5.0 | 8.3 | 5.1 |
| **Random** | PointReach | $\mathbf{29.26}_{\pm0.04}$ | $29.21_{\pm0.08}$ | $\mathbf{29.31}_{\pm0.04}$ | $23.96_{\pm0.93}$ | $25.76_{\pm0.96}$ | $17.74_{\pm1.84}$ | $25.55_{\pm0.57}$ | $10.12_{\pm0.72}$ | $22.65_{\pm1.57}$ |
| | PointRooms | $24.80_{\pm0.36}$ | $24.07_{\pm0.19}$ | $\mathbf{25.16}_{\pm0.59}$ | $18.09_{\pm4.13}$ | $19.41_{\pm1.01}$ | $14.69_{\pm2.51}$ | $19.10_{\pm1.39}$ | $5.76_{\pm2.99}$ | $20.88_{\pm0.96}$ |
| | Reacher | $21.09_{\pm0.65}$ | $16.65_{\pm0.48}$ | $22.24_{\pm0.54}$ | $\mathbf{25.10}_{\pm0.68}$ | $22.98_{\pm0.91}$ | $10.62_{\pm2.30}$ | $\mathbf{23.70}_{\pm0.62}$ | $4.74_{\pm0.36}$ | $6.06_{\pm0.84}$ |
| | SawyerReach | $26.70_{\pm0.14}$ | $25.46_{\pm0.12}$ | $\mathbf{26.86}_{\pm0.07}$ | $19.48_{\pm1.39}$ | $21.32_{\pm1.40}$ | $8.78_{\pm2.59}$ | $25.29_{\pm0.35}$ | $3.46_{\pm0.86}$ | $2.84_{\pm0.05}$ |
| | SawyerDoor | $19.05_{\pm0.66}$ | $18.26_{\pm1.18}$ | $\mathbf{21.69}_{\pm2.36}$ | $\mathbf{20.69}_{\pm2.14}$ | $19.58_{\pm3.55}$ | $12.47_{\pm3.08}$ | $10.82_{\pm1.67}$ | $7.92_{\pm0.86}$ | $14.77_{\pm0.51}$ |
| | FetchReach | $\mathbf{30.42}_{\pm0.04}$ | $30.38_{\pm0.06}$ | $\mathbf{30.42}_{\pm0.04}$ | $28.34_{\pm0.98}$ | $27.94_{\pm0.30}$ | $18.96_{\pm1.77}$ | $27.11_{\pm0.22}$ | $1.71_{\pm0.77}$ | $1.21_{\pm0.46}$ |
| | FetchPush | $5.21_{\pm0.43}$ | $5.08_{\pm0.32}$ | $\mathbf{7.22}_{\pm0.35}$ | $6.99_{\pm1.27}$ | $5.35_{\pm3.36}$ | $4.22_{\pm2.19}$ | $4.53_{\pm1.94}$ | $4.49_{\pm1.34}$ | $5.35_{\pm0.23}$ |
| | FetchPick | $3.75_{\pm0.18}$ | $3.02_{\pm0.16}$ | $\mathbf{4.36}_{\pm0.19}$ | $\mathbf{3.81}_{\pm3.71}$ | $1.87_{\pm1.59}$ | $0.81_{\pm0.82}$ | $3.08_{\pm1.35}$ | $2.16_{\pm0.75}$ | $2.17_{\pm0.18}$ |
| | FetchSlide | $\mathbf{2.67}_{\pm0.35}$ | $0.00_{\pm0.00}$ | $\mathbf{3.15}_{\pm0.14}$ | $1.32_{\pm1.22}$ | $1.04_{\pm0.98}$ | $0.24_{\pm0.27}$ | $1.12_{\pm0.39}$ | $1.31_{\pm0.52}$ | $0.00_{\pm0.00}$ |
| | HandReach | $\mathbf{14.89}_{\pm2.54}$ | $0.00_{\pm0.00}$ | $\mathbf{17.61}_{\pm3.06}$ | $0.08_{\pm0.07}$ | $2.54_{\pm1.42}$ | $1.41_{\pm0.51}$ | $0.00_{\pm0.00}$ | $0.00_{\pm0.00}$ | $0.00_{\pm0.00}$ |
| | Average Rank | **2.8** | 4.8 | **1.3** | 3.8 | 4.5 | 7.3 | 5.3 | 7.7 | 6.7 |

Table 7: Success rates, averaged over 10 seeds.

| | Task Name | Ours | | | Offline GCRL | | | | Diffusion-based | |
|---|---|---|---|---|---|---|---|---|---|---|
| | | Merlin | Merlin-P | Merlin-NP | GoFAR | WGCSL | GCSL | AM | DD | g-DQL |
| **Expert** | PointReach | $\mathbf{1.00}_{\pm0.00}$ | $\mathbf{1.00}_{\pm0.00}$ | $\mathbf{1.00}_{\pm0.00}$ | $\mathbf{1.00}_{\pm0.00}$ | $\mathbf{1.00}_{\pm0.00}$ | $\mathbf{1.00}_{\pm0.00}$ | $\mathbf{1.00}_{\pm0.00}$ | $0.40_{\pm0.00}$ | $\mathbf{1.00}_{\pm0.00}$ |
| | PointRooms | $0.91_{\pm0.16}$ | $0.89_{\pm0.04}$ | $\mathbf{0.94}_{\pm0.01}$ | $0.82_{\pm0.04}$ | $0.82_{\pm0.04}$ | $0.79_{\pm0.6}$ | $0.87_{\pm0.05}$ | $0.27_{\pm0.17}$ | $\mathbf{1.00}_{\pm0.00}$ |
| | Reacher | $\mathbf{1.00}_{\pm0.00}$ | $\mathbf{1.00}_{\pm0.00}$ | $\mathbf{1.00}_{\pm0.00}$ | $\mathbf{1.00}_{\pm0.00}$ | $\mathbf{1.00}_{\pm0.00}$ | $\mathbf{1.00}_{\pm0.00}$ | $\mathbf{1.00}_{\pm0.00}$ | $0.20_{\pm0.00}$ | $\mathbf{1.00}_{\pm0.00}$ |
| | SawyerReach | $\mathbf{1.00}_{\pm0.00}$ | $\mathbf{1.00}_{\pm0.00}$ | $\mathbf{1.00}_{\pm0.00}$ | $\mathbf{1.00}_{\pm0.00}$ | $\mathbf{1.00}_{\pm0.00}$ | $\mathbf{1.00}_{\pm0.00}$ | $\mathbf{1.00}_{\pm0.00}$ | $0.13_{\pm0.05}$ | $\mathbf{1.00}_{\pm0.00}$ |
| | SawyerDoor | $\mathbf{0.95}_{\pm0.08}$ | $0.92_{\pm0.08}$ | $\mathbf{0.94}_{\pm0.11}$ | $0.82_{\pm0.12}$ | $0.86_{\pm0.15}$ | $0.84_{\pm0.16}$ | $0.92_{\pm0.12}$ | $0.20_{\pm0.00}$ | $\mathbf{0.94}_{\pm0.12}$ |
| | FetchReach | $\mathbf{1.00}_{\pm0.00}$ | $\mathbf{1.00}_{\pm0.00}$ | $\mathbf{1.00}_{\pm0.00}$ | $\mathbf{1.00}_{\pm0.00}$ | $\mathbf{1.00}_{\pm0.00}$ | $0.98_{\pm0.00}$ | $\mathbf{1.00}_{\pm0.00}$ | $0.07_{\pm0.04}$ | $\mathbf{1.00}_{\pm0.00}$ |
| | FetchPush | $0.95_{\pm0.05}$ | $0.07_{\pm0.03}$ | $\mathbf{0.96}_{\pm0.04}$ | $\mathbf{0.96}_{\pm0.04}$ | $0.95_{\pm0.04}$ | $0.88_{\pm0.09}$ | $0.90_{\pm0.08}$ | $0.17_{\pm0.09}$ | $0.89_{\pm0.11}$ |
| | FetchPick | $\mathbf{0.92}_{\pm0.03}$ | $0.05_{\pm0.03}$ | $\mathbf{0.96}_{\pm0.06}$ | $0.78_{\pm0.04}$ | $0.76_{\pm0.07}$ | $0.64_{\pm0.09}$ | $0.69_{\pm0.16}$ | $0.07_{\pm0.07}$ | $0.91_{\pm0.07}$ |
| | FetchSlide | $0.20_{\pm0.04}$ | $0.00_{\pm0.00}$ | $0.25_{\pm0.07}$ | $0.28_{\pm0.09}$ | $\mathbf{0.42}_{\pm0.14}$ | $0.22_{\pm0.14}$ | $\mathbf{0.32}_{\pm0.12}$ | $0.10_{\pm0.08}$ | $0.05_{\pm0.04}$ |
| | HandReach | $\mathbf{0.78}_{\pm0.04}$ | $0.00_{\pm0.00}$ | $\mathbf{0.85}_{\pm0.02}$ | $0.54_{\pm0.23}$ | $0.68_{\pm0.19}$ | $0.03_{\pm0.05}$ | $0.00_{\pm0.00}$ | $0.00_{\pm0.00}$ | $0.00_{\pm0.00}$ |
| | Average Rank | **2.1** | 4.5 | **1.5** | 3.0 | 2.8 | 5.0 | 3.2 | 8.3 | 3.0 |
| **Random** | PointReach | $\mathbf{1.00}_{\pm0.00}$ | $\mathbf{1.00}_{\pm0.00}$ | $\mathbf{1.00}_{\pm0.00}$ | $\mathbf{1.00}_{\pm0.00}$ | $\mathbf{1.00}_{\pm0.00}$ | $\mathbf{1.00}_{\pm0.00}$ | $\mathbf{1.00}_{\pm0.00}$ | $0.40_{\pm0.00}$ | $0.95_{\pm0.06}$ |
| | PointRooms | $\mathbf{0.89}_{\pm0.02}$ | $0.84_{\pm0.03}$ | $\mathbf{0.92}_{\pm0.02}$ | $0.78_{\pm0.11}$ | $0.83_{\pm0.08}$ | $0.77_{\pm0.11}$ | $0.70_{\pm0.16}$ | $0.27_{\pm0.17}$ | $0.82_{\pm0.08}$ |
| | Reacher | $0.98_{\pm0.02}$ | $0.89_{\pm0.05}$ | $\mathbf{1.00}_{\pm0.00}$ | $0.98_{\pm0.03}$ | $\mathbf{1.00}_{\pm0.00}$ | $0.80_{\pm0.06}$ | $\mathbf{1.00}_{\pm0.00}$ | $0.23_{\pm0.05}$ | $0.15_{\pm0.05}$ |
| | SawyerReach | $\mathbf{1.00}_{\pm0.00}$ | $0.98_{\pm0.02}$ | $\mathbf{1.00}_{\pm0.00}$ | $0.92_{\pm0.07}$ | $\mathbf{1.00}_{\pm0.00}$ | $0.91_{\pm0.09}$ | $\mathbf{1.00}_{\pm0.00}$ | $0.13_{\pm0.05}$ | $0.10_{\pm0.03}$ |
| | SawyerDoor | $\mathbf{0.57}_{\pm0.03}$ | $0.49_{\pm0.08}$ | $\mathbf{0.59}_{\pm0.05}$ | $0.46_{\pm0.19}$ | $0.48_{\pm0.17}$ | $0.44_{\pm0.16}$ | $0.26_{\pm0.09}$ | $0.20_{\pm0.00}$ | $0.35_{\pm0.09}$ |
| | FetchReach | $\mathbf{1.00}_{\pm0.00}$ | $\mathbf{1.00}_{\pm0.00}$ | $\mathbf{1.00}_{\pm0.00}$ | $\mathbf{1.00}_{\pm0.00}$ | $\mathbf{1.00}_{\pm0.00}$ | $0.96_{\pm0.05}$ | $\mathbf{1.00}_{\pm0.00}$ | $0.07_{\pm0.04}$ | $0.00_{\pm0.00}$ |
| | FetchPush | $0.20_{\pm0.06}$ | $0.17_{\pm0.01}$ | $\mathbf{0.24}_{\pm0.09}$ | $\mathbf{0.22}_{\pm0.04}$ | $0.14_{\pm0.10}$ | $0.20_{\pm0.11}$ | $0.13_{\pm0.09}$ | $0.13_{\pm0.05}$ | $0.17_{\pm0.04}$ |
| | FetchPick | $\mathbf{0.12}_{\pm0.01}$ | $0.09_{\pm0.01}$ | $\mathbf{0.18}_{\pm0.01}$ | $\mathbf{0.12}_{\pm0.11}$ | $0.08_{\pm0.07}$ | $0.06_{\pm0.08}$ | $0.10_{\pm0.02}$ | $0.07_{\pm0.05}$ | $0.09_{\pm0.02}$ |
| | FetchSlide | $\mathbf{0.11}_{\pm0.02}$ | $0.00_{\pm0.00}$ | $\mathbf{0.20}_{\pm0.04}$ | $0.10_{\pm0.06}$ | $0.04_{\pm0.08}$ | $0.06_{\pm0.08}$ | $0.07_{\pm0.04}$ | $0.07_{\pm0.05}$ | $0.00_{\pm0.00}$ |
| | HandReach | $\mathbf{0.49}_{\pm0.05}$ | $0.00_{\pm0.00}$ | $\mathbf{0.62}_{\pm0.07}$ | $0.00_{\pm0.00}$ | $0.12_{\pm0.07}$ | $0.04_{\pm0.04}$ | $0.00_{\pm0.00}$ | $0.00_{\pm0.00}$ | $0.00_{\pm0.00}$ |
| | Average Rank | **2.0** | 4.1 | **1.0** | 3.5 | 3.5 | 5.6 | 4.1 | 7.6 | 6.9 |

