# OpenReview forum: "Learning to Reach Goals via Diffusion"
_ICLR.cc/2024/Conference — ICLR 2024 Conference Withdrawn Submission_

### Official Review · Reviewer_euBm · 2023-10-29

**Soundness:** 3 good
**Presentation:** 2 fair
**Contribution:** 2 fair
**Rating:** 5
**Confidence:** 4

**Summary:**

This paper presents a new approach called Merlin for goal-conditioned reinforcement learning, which is inspired from diffusion models. The authors introduce a new approach to construct a "forward process" by stitching trajectories if there are two states from different trajectories are close. The forward process outputs a augmented dataset, and authors propose to learn the corresponding backward process. They validate their method on offline goal-reaching tasks and show competitive results with state-of-the-art methods. Overall, the paper proposes a new class of goal-conditioned RL algorithms,

**Strengths:**

1. I like the high level idea of this work which is inspired from diffusion models: constructing a simple forward process to enlarge the training set by injecting noise, and learning the reverse process. Specifically, they use the Nearest-neighbor Trajectory Stitching to generate more data. The algorithm is somewhat novel and might work well on some tasks.

2. Competitive results: The authors validate their approach on offline goal-reaching tasks and show competitive results with state-of-the-art methods. This demonstrates the effectiveness of their approach and its potential for real-world applications.

**Weaknesses:**

1. Weak theoretical justification: diffusion models enjoy strong theoretical foundations, the forward and the backward process are proven to share the same marginal distribution. However, it is not clear to me whether the backward process of  Nearest-neighbor Trajectory Stitching still has similar theoretical guarantees.

2. Limited range of applications: Nearest-neighbor Trajectory Stitching seems to be designed for some specific applications. The generalizability remains unclear.

3. Misleading title: Diffusion models have a relatively clear definition now. While there are "forward" and "backward" processes in this paper, this algorithm does not fall into the class of diffusion models.

**Questions:**

See weaknesses.

---

> ### Author Response · Authors · 2023-11-22
> **Response by Authors**
>
> Thank you for the valuable feedback! We respond to your individual queries below; we hope the changes to the paper and the responses address your concerns.
>
> > Weak theoretical justification: diffusion models enjoy strong theoretical foundations, the forward and the backward process are proven to share the same marginal distribution. However, it is not clear to me whether the backward process of Nearest-neighbor Trajectory Stitching still has similar theoretical guarantees.
>
> Interesting question: our theoretical justification given in theorem 5.1 relates likelihood maximization to behaviour cloning for a given dataset. While trajectory stitching is a heuristic data argumentation process, it only results in a different augmented dataset. Our theorem still applies in the sense that for the augmented dataset, likelihood maximization is still related to behaviour cloning. In short, our theorem justifies using a diffusion-like process for goal-conditioned RL, starting from a given dataset. Introducing data augmentation and model into the picture is separated since they effectively change/update the dataset.
>
> > Limited range of applications: Nearest-neighbor Trajectory Stitching seems to be designed for some specific applications. The generalizability remains unclear.
>
> The trajectory stitching operation can be used with any suitable distance metric. In particular, for images and cases where such a metric is not obvious, the trajectory stitching can be performed in a latent space by learning a mapping from the state space to a low-dimensional space using any viable representation learning method. This commonly used technique has been effective for dealing with high-dimensional and complex data (such as images). We have also added a discussion in Section 7 highlighting this point.
>
> > Misleading title: Diffusion models have a relatively clear definition now. While there are "forward" and "backward" processes in this paper, this algorithm does not fall into the class of diffusion models.
>
> Denoising diffusion probabilistic models (DDPM) [1], use Gaussian noise for the forward process, but they represent “one” possible type of diffusion model. In its general form [2], a diffusion model requires a forward noising Markov chain and a denoising model. In RL, the forward process comprises taking actions starting from the goal, which leads to new states different from the goal. Since we are dealing with an MDP, the forward and backward chains thus generated are Markov chains by definition. Our approach is indeed a diffusion process, and we discuss the connection and differences with DDPM in Appendix C. Also, Theorem 5.1 is reminiscent of a related  proof in [2].
>
> [1] Ho, Jonathan, Ajay Jain, and Pieter Abbeel. "Denoising diffusion probabilistic models." Advances in neural information processing systems 33 (2020): 6840-6851.
>
> [2] Sohl-Dickstein, Jascha, Eric Weiss, Niru Maheswaranathan, and Surya Ganguli. "Deep unsupervised learning using nonequilibrium thermodynamics." In International conference on machine learning, pp. 2256-2265. PMLR, 2015.

---

### Official Review · Reviewer_5ke8 · 2023-10-31

**Soundness:** 2 fair
**Presentation:** 2 fair
**Contribution:** 2 fair
**Rating:** 3
**Confidence:** 3

**Summary:**

This paper models the offline GCRL problem in offline data in a diffusion process-like paradigm called merlin. The authors consider three choices for the noise model to replace Gaussian noise in diffusion including reverse play from the buffer, reverse dynamics model, and a novel non-parametric trajectory stitching. This is an improved behavioural cloning paradigm without the need to learn an additional value function, which achieves excellent results in offline control tasks.

**Strengths:**

1.Novel perspective of framing goal-reaching as a diffusion process.
2.Trajectory stitching technique seems useful for generating diverse state-goal pairs from offline data.
3.Strong empirical results on offline goal-reaching tasks compared to prior methods.

**Weaknesses:**

1.Although the paper seems to describe a feasible diffusion-like process to model the GCRL problem, I think merlin is essentially a variant of constrained GCSL. From this perspective, merlin has only limited novelty. Start with the cleanest method, merlin build policy upon $s, g, h$ instead of $s, g$ by GCSL. Although the merlin shows better results in the motivation example, I think it's because of the inclusion of a more stable time guide.
2.I observe that Merlin-NP and Merlin-P show better results in the experiments, but they can be considered as GCSL + temporal constraints + reverse dynamics model (Wang et al.) + trajectory stitching (a commonly used data augmentation method in OfflineRL). These other components can be easily combined with the universal GCRL approach, so the performance gains are no surprise.
3.The approach seems sensitive to hyperparameters like time horizon and hindsight ratio. I'm not sure that good performance comes from hyperparameter tuning.

**Questions:**

1.What metric and distance threshold works best for the trajectory stitching? Is there a principled way to set this?
2.In appendix table 5, I have observed that there is not much difference in success rate between Merlin and DQL, GCSL and other methods, whereas there is a bigger difference using reward metric, why is that? Success rate should be a common metric for evaluating a GCRL algorithm.

Overall this paper proposes interesting ideas for offline goal-conditioned RL as diffusion process. The empirical results are strong but there are some open questions (see above weakness and questions). Addressing some of the weaknesses and questions raised would strengthen the paper further. I think the central problem is that the article overclaimed the design of the approach to solving the GCRL problem by a diffusion process and I vote reject for current version.

---

> ### Author Response · Authors · 2023-11-22
> **Response by Authors**
>
> Thank you for the valuable feedback! We respond to your individual queries below, we hope the changes to the paper and the responses address your concerns.
>
> > Although the paper seems to describe a feasible diffusion-like process to model the GCRL problem, I think merlin is essentially a variant of constrained GCSL. From this perspective, merlin has only limited novelty. Start with the cleanest method, merlin build policy upon s,g,h instead of s,g by GCSL. Although the merlin shows better results in the motivation example, I think it's because of the inclusion of a more stable time guide.
>
> We extensively discuss and contrast Merlin and GCSL in Section 6. As stated, GCSL also allows for conditioning on the time horizon, but its authors did not find any significant difference with and without the time horizon. In Merlin, the inclusion of time is coupled with variance prediction (see Figure 3), and is one reason Merlin is superior to GCSL. This is also what practically sets apart score-based models and diffusion models in the generative modeling task and, therefore, should not be deemed insignificant. Also, as discussed in Section 6, the forward view of GCSL and the backward view of Merlin can result in very different outcomes. Consider the model-based approach: a forward dynamics model generates trajectories without guarantees on the distribution over the goal state. In contrast, in a reverse dynamics model, one has control over this distribution.
>
> > I observe that Merlin-NP and Merlin-P show better results in the experiments, but they can be considered as GCSL + temporal constraints + reverse dynamics model (Wang et al.) + trajectory stitching (a commonly used data augmentation method in OfflineRL). These other components can be easily combined with the universal GCRL approach, so the performance gains are no surprise.
>
> We are unaware of any prior offline RL method that uses trajectory stitching for data augmentation. Could the reviewer please provide a reference? While there is a parallel between GCSL and Merlin, the diffusion perspective only applies to Merlin, and as explained in response to the previous questions, this makes a significant difference, both in theory (e.g., theorem 5.1) and practice (experimental results). We also ran GCSL+ forward trajectory stitching to show that it remains inferior to Merlin-NP, the results are presented in Appendix G.
>
> > The approach seems sensitive to hyperparameters like time horizon and hindsight ratio. I'm not sure that good performance comes from hyperparameter tuning.
>
> While we agree and have discussed this in sections 6 and 7, performance sensitivity to hindsight ratio is not unique to Merlin, and one observes the same for most of the offline GCRL methods. For each baseline, we used the best-performing hindsight ratio as reported in their original works. For transparency, we provide an ablation of these hyper-parameters in Section 6.1. More generally, RL methods are notorious for being sensitive to hyper-parameters, and most methods require some hyper-parameter tuning to be effective.
>
> > What metric and distance threshold works best for the trajectory stitching? Is there a principled way to set this?
>
> The choice of metric and the distance threshold is discussed in Appendix D.3..
>
> > In appendix table 5, I have observed that there is not much difference in success rate between Merlin and DQL, GCSL and other methods, whereas there is a bigger difference using reward metric, why is that? Success rate should be a common metric for evaluating a GCRL algorithm.
>
> The discounted return takes into account how fast the agent reaches the goal and whether it stays in the goal region thereafter. The success rate is important but does not provide the full picture - multiple methods may successfully solve the task but may take a different numbers of steps to reach the goal, and some may deviate from the goal after successfully reaching it. This is briefly discussed in Section 6.

---

### Official Review · Reviewer_6Zbj · 2023-10-31

**Soundness:** 2 fair
**Presentation:** 2 fair
**Contribution:** 2 fair
**Rating:** 5
**Confidence:** 3

**Summary:**

The paper proposes a diffusion based method for goal-conditioned Reinforcement Learning. It is assumed that a dataset of offline demonstrations is given (which also indicate a goal variable g). This dataset is then used to train a diffusion-model-based policy. The idea is to inverse-diffuse the current sample to eventually arrive at the goal point. Hence, the noising process is in state space, where the idea is that the goal state is continuously noised (generating a reversed trajectory).

**Strengths:**

- The problem setting is interesting
- The figures are nice and intuitively explain the ideas presented in the paper
- The analogies to behavior cloning are interesting
- Reduction in need for denoising steps is beneficial

**Weaknesses:**

- The introduction could be improved by making the exact problem setting more clear from the beginning
- The nearest neighbor based approach makes the assumtion that close states are connected/ can be accessd from each other, this should be discussed. This could also be evaluated by designing a more complex toy environment based on the environment in Figure 2.
- The related work description of Janner et all is not exactly correct, as it is not full trajectories that are noised but just trajectory segments
- A comparison to the related works such as [A] would be appreciated
- An ablation on trajectory stitching is only implicitly done (by defining different algorithms)
- A motivation for the goal-conditioned problem setting (instead of starting with just a single goal setting) would be beneficial

-.

- The description of the method is rather confusing. First, it is explained that the trajectory is denoised, which would result in a denoising of states. However, in the following sections, suddenly the action is denoised (see section 4.1)
- Is the policy a diffusion model? It is mentioned that BC is performed at the end of section 5.2.
- The paper would definitely benefit from an algorithm description of the method
- It appears that the dataset extension through trajectory stitching is not performed for the baseline methods, which makes the comparison unfair
- The fact that methods based on inverse dynamics model approaches did not work weakens the method, as trajectoriy stitching has obv. downsides and likely only works in state spaces that resemble physical environments


Related work:

[1] "Goal-Conditioned Imitation Learning using Score-based Diffusion Policies", Reuss et al. 2023

**Questions:**

See weaknesses

---

> ### Author Response · Authors · 2023-11-22
> **Response by Authors (1/2)**
>
> Thank you for the valuable feedback! We respond to your individual queries below, we hope the changes to the paper and the responses address your concerns.
>
> > The introduction could be improved by making the exact problem setting more clear from the beginning
>
> We have expanded upon the explanation of goal-conditioned RL in the introduction to make the problem setting more clear. The formal introduction in Section 3.2 specifies the notation and the problem setting used throughout the paper. We hope the changes help improve the clarity of the paper.
>
> > The nearest neighbor-based approach makes the assumption that close states are connected/ can be accessed from each other; this should be discussed. This could also be evaluated by designing a more complex toy environment based on the environment in Figure 2.
>
> We have added experiments in Appendix B using a more complex toy environment, which adds numerous walls that the agent must learn to navigate around. We observe that Merlin successfully learns to navigate around the walls in contrast to GCSL. For more complex state spaces (for example, images, where nearby states in pixel-space may not be directly accessible), the trajectory stitching operation can be performed in a learned latent space by learning a mapping from the state space to a low-dimensional space using any viable representation learning method. This commonly used technique has been effective for dealing with high-dimensional and complex data. We have also added a discussion in Section 7 highlighting this point.
>
> > The related work description of Janner et all is not exactly correct, as it is not full trajectories that are noised but just trajectory segments
>
> We have corrected the statement in the related works section; thank you for pointing this out.
>
> > A comparison to the related works such as [A] would be appreciated
>
> Due to the limited time period of this discussion phase, we were unable to produce full results for this work; we shall work on it for the final version of the paper. However, note that according to the ICLR guidelines, it is not expected to compare with works that are published after 28 May, 2023.
>
> > An ablation on trajectory stitching is only implicitly done (by defining different algorithms)
>
> Could the reviewer please clarify? We report the performance with and without trajectory stitching, which demonstrates the effect of this technique on the performance. Could the reviewer elaborate on the suggested experimental setup?
>
> > A motivation for the goal-conditioned problem setting (instead of starting with just a single goal setting) would be beneficial
>
> We assume the reviewer is referring to the experiments in Section 4.1. We have added results for the multi-goal setting in Appendix B, where the forward diffusion is performed starting from several goals. During evaluation, the policy is conditioned on one particular goal. Thank you for the suggestion.
>
> > The description of the method is rather confusing. First, it is explained that the trajectory is denoised, which would result in a denoising of states. However, in the following sections, suddenly the action is denoised (see section 4.1)
>
> The diffusion process in Merlin is over states. The forward process comprises of taking actions starting from the goal, which can be seen as equivalent to adding Gaussian noise in diffusion models, which leads to new states different from the goal. The policy, which is akin to the score function (as explained in Section 4.1), produces the action to reach the appropriate states. In this sense, the action is analogous to the predicted noise in diffusion models, which serves to “denoise” the states towards the goal. We have added a sentence in Section 4.1 to improve clarity.
>
> > Is the policy a diffusion model? It is mentioned that BC is performed at the end of section 5.2.
>
> Yes, see the previous answer and Section 4.1 for an explanation. Theorem 5.1 proves that maximizing the ELBO in the diffusion process becomes equivalent to behavior cloning in the offline setting.
>
> > The paper would definitely benefit from an algorithm description of the method
>
> We have added formal algorithms for the reverse model rollout (for Merlin-P) and for Merlin, including all its variations in Appendix D. If the reviewer finds it useful, page limit permitting, we can move it to the main body of the paper.

---

> ### Author Response · Authors · 2023-11-22
> **Response by Authors (2/2)**
>
> > It appears that the dataset extension through trajectory stitching is not performed for the baseline methods, which makes the comparison unfair.
>
> The trajectory stitching is a contribution of our work, which was proposed to construct reverse trajectories for the forward diffusion process. Since it relies on reverse trajectories in its original form, the technique can’t be applied to other methods, which we used as baselines as proposed in their original publications. However, we applied a modified version of this to GCSL, to demonstrate that it can benefit other offline RL methods. The results are presented in Appendix G. As expected, it improves GCSL, but the resulting GCSL algorithm remains inferior to Merlin-NP.
>
> > The fact that methods based on inverse dynamics model approaches did not work weakens the method, as trajectoriy stitching has obv. downsides and likely only works in state spaces that resemble physical environments
>
> It is true that Merlin-P, which uses the reverse dynamics model to generate the forward diffusion trajectories, does not work very well for some of the tasks with high-dimensional state space. We This is a well-known failure of model-based RL, due to compounding model error over multiple time steps especially in higher dimensions, and is briefly discusses in Section 6. However, Merlin is a general framework for approaching GCRL problems inspired by diffusion models, and using the reverse dynamics model is “one” way to specify the forward process. For more complex environments, one could apply trajectory stitching in a learned latent space using any viable representation learning method. This commonly used technique has been effective for dealing with high-dimensional and complex data.

---

> ### Comment · Reviewer_6Zbj · 2023-11-22
>
> Thank you for the provided clarifications and the additional experiments.
>
> I am not sure I would see nearest-neighbor trajectory stitching as a novel contribution to the field. See e.g. https://arxiv.org/pdf/2204.12026.pdf or https://link.springer.com/article/10.1007/s11227-019-02813-w. While the stitching algorithm proposed in this work may have some technical differences from previous works, I would not consider it a significant contribution. Further, as the authors pointed out themselves, it does not generalize to problems with more complex state spaces where a "connection" between nearest neighbors cannot be assumed to be given.
>
> However, the MERLIN algorithm itself is a novel contribution and the additional results provided in Appendix G suggest that it is performant. One thing I am still confused about: The work assumes that actions can be simulated as random noise added to the states. Does this assume that the state and action space are of the same dimension? Or do you consider the Gaussian noise as a meta-action (i.e. a change in the state), which is then passed to a low-level controller that executes the according control action?
> Further, assuming that any state is connected to the same state with added Gaussian noise similarly assumes that these states are connected. Specifically, this assumes that any state is connected to a noised version of itself, through taking a single action. This appears as a very strong assumption which I find is not discussed in much detail in the paper. Hence, I am curious how this method could generalize to state spaces where this assumption does not hold (e.g. image inputs).

---

> > ### Author Response · Authors · 2023-11-22
> > **Response by Authors**
> >
> > Thank you for your response.
> >
> > The trajectory stitching is one way we propose to build the forward diffusion chain for applying Merlin. In our experiments, we test Merlin-NP on ten tasks with different state spaces, including some tasks (such as PointRooms) where nearby states need not necessarily be connected, and the trajectory stitching helps improve performance in all of these cases. Further, as stated in our response, this technique can be applied in a learned latent space for more complex state space.
> >
> > Regarding the second point, there seems to be a misunderstanding. The diffusion process uses the environment transitions in order to produce the next state given a state and action, as defined in the MDP (see Section 5.2). Since the reverse dynamics are not known, we propose learning a reverse dynamics model (Merlin-P) or trajectory stitching (Merlin-NP) to produce possible forward diffusion chains. In fact, apart from the 2D navigation example, all of the tasks considered in our experiments have different state and action space dimensions. Hence, there is no assumption that states are connected and the method can easily be applied to image inputs as well.

---

### Official Review · Reviewer_i457 · 2023-11-05

**Soundness:** 2 fair
**Presentation:** 3 good
**Contribution:** 2 fair
**Rating:** 5
**Confidence:** 3

**Summary:**

This presents a method for sequential decision making with diffusion. It frames sequential decision making as the reverse process in diffusion. In this case the initial state is “noise” and the final state is the result of denoising. For a particular goal state the policy will “denoise” the initial state. An additional contribution of this work is their “trajectory stitching method”. If there are states that are nearby to one another in two different trajectories then the dataset can be augmented by concatenation of trajectory segments (making sure to relabel the goal state for the swapped trajectories).

**Strengths:**

Interesting dataset augmentation technique that might improve performance on some control tasks.

**Weaknesses:**

The trajectory stitching method is only usable if distance between two states can be defined. What if states are observed via images? Additionally what if distance between states is not indicative of their relation to one another in a sequential process. What if there are discontinuities in states?

Transition from 3.2 to 4 is abrupt. No additional information on issues with offline reinforcement learning.

GCSL seems to be a very important concept which is used as a baseline algorithm in this paper. Yet there is no description of it in related work. How is GCSL different from GCRL?

Figure 7 is referenced in the main text but appears in the appendix.

**Questions:**

is the method applicable with partially observable states e.g., images?

---

> ### Author Response · Authors · 2023-11-22
> **Response by Authors**
>
> Thank you for the valuable feedback! We respond to your individual queries below, we hope the changes to the paper and the responses address your concerns.
>
> > The trajectory stitching method is only usable if distance between two states can be defined. What if states are observed via images? Additionally what if distance between states is not indicative of their relation to one another in a sequential process. What if there are discontinuities in states?
>
> Interesting point! Indeed trajectory stitching operation can be used with any suitable distance metric. In particular, for images and cases where such a metric is not obvious, the trajectory stitching can be performed in a latent space by learning a mapping from the state space to a low-dimensional space using any viable representation learning method. This commonly used technique has been effective for dealing with high-dimensional and complex data (such as images). We have added a discussion in Section 7 highlighting this point.
>
> > Transition from 3.2 to 4 is abrupt. No additional information on issues with offline reinforcement learning.
>
> In Section 1, we discuss the inaccuracies in value estimation, especially for out-of-distribution actions, which can cause policies to diverge. The second issue is that the static dataset provides limited state-goal pairs for training policies. Since having such a discussion in technical details is not feasible given our constraints, we do this at the high level in Section 1, and reserve Section 3.2   for basic notation and fundamentals of offline RL which is used in the subsequent sections.
>
> > GCSL seems to be a very important concept which is used as a baseline algorithm in this paper. Yet there is no description of it in related work. How is GCSL different from GCRL?
>
> GCRL is a particular framework of RL where the objective is defined in terms of the desired goal that the policy is expected to achieve. GCSL [1] is one proposed method to solve the GCRL problem. We have described GCSL in the related work section under the paragraph ‘Offline Goal-Conditioned Reinforcement Learning’, and again in the Experiments section under the paragraph ‘Algorithms.’ We have expanded the introduction to make the setting of interest more clear.
>
> > Figure 7 is referenced in the main text but appears in the appendix.
>
> This was a typing error, it should refer to Figure 4, which shows the generalization of Merlin to new goals. Thank you for pointing this out.
>
> > is the method applicable with partially observable states e.g., images?
>
> This is an interesting suggestion and is one of the possible avenues for future work. In principle, it should work for POMDPs if we use recurrent networks for the policy and the reverse dynamics model (in the case of Merlin-P). Using a recurrent network to keep track of the entire history of partially observed states has been shown to be effective [2,3].
>
>
> [1] Ghosh, Dibya, Abhishek Gupta, Ashwin Reddy, Justin Fu, Coline Manon Devin, Benjamin Eysenbach, and Sergey Levine. "Learning to Reach Goals via Iterated Supervised Learning." In International Conference on Learning Representations. 2020.
>
> [2] Hausknecht, Matthew, and Peter Stone. "Deep recurrent q-learning for partially observable mdps." In 2015 aaai fall symposium series. 2015.
>
> [3] Ni, Tianwei, Benjamin Eysenbach, and Ruslan Salakhutdinov. "Recurrent Model-Free RL Can Be a Strong Baseline for Many POMDPs." In International Conference on Machine Learning, pp. 16691-16723. PMLR, 2022.

---

### Author Response · Authors · 2023-11-22
**Summary of major changes**

We thank all reviewers for their thorough reviews and constructive comments. We address each reviewer’s concerns individually; here, we provide a brief summary of major modifications to the paper, which are highlighted in red.
- Updated proof of Theorem 5.1 in Appendix A to cover stochastic transitions.
- Added more illustrative experiments, including multi-goal settings and a more complex navigation task in Appendix B.
- Added formal algorithm for Merlin and trajectory generation using reverse model (for Merlin-P) in Appendix D.1
- Added quantitative comparison of training and inference time for all baselines in Appendix D.5, showing Merlin is orders of magnitude faster than other diffusion-based methods.
- Added experimental results of applying a modified version of the proposed trajectory stitching technique to GCSL in Appendix G, demonstrating that it can benefit other methods as well.